# Paleolatitude.org 3.0: A calculator for paleoclimate and paleobiology studies based on a new global paleogeography model

Douwe J. J. van Hinsbergen[1]*, Bram Vaes[2,3], Lydian M. Boschman[1], Nalan Lom[4],
Suzanna H. A. van de Lagemaat[1], Eldert L. Advokaat[1], Sanne de Baar[1],
Menno R. T. Fraters[5], Joren Paridaens[6], Emilia B. Jarochowska[1]

**1** Department of Earth Sciences, Utrecht University, Utrecht, The Netherlands, **2** Department of Earth and Environmental Sciences, Università degli Studi di Milano-Bicocca, Milan, Italy, **3** Aix-Marseille Université, CNRS, IRD, INRAE, CEREGE, Aix-en-Provence, France, **4** Department of Earth and Environmental Sciences, University of Central Asia, Khorog, Tajikistan, **5** University of Graz, Department of Earth Sciences, Graz, Austria, **6** 9de Online, Utrecht, The Netherlands

* d.j.j.vanhinsbergen@uu.nl

## Abstract

Paleogeography, and particularly the paleolatitude, provides key context in the interpretation of paleoclimatic and paleobiological data but these fields are typically studied by scientists in different disciplines. To facilitate communication between these disciplines, a decade ago the online Paleolatitude.org calculator was developed. This provided for any coordinate on stable tectonic plates a paleolatitude estimate for any chosen Phanerozoic time interval, including an uncertainty that includes paleogeographic uncertainty and age uncertainty of a sample/fossil. Here, we provide a major update to this tool. First, we include in the calculator the first global paleogeographic model, including GPlates reconstruction files, back to 320 Ma that also restores paleogeographic units that are now thrusted over each other in orogenic (mountain) belts. Second, we include a recent, more precise paleomagnetic reference frame with updated statistical procedures, and provide the first update of its underlying database. Third, we introduce a new online interface with an easy-to-use tool with a batch option, and data and graph export functions. Finally, we illustrate differences with previous reconstructions and show an application by calculating a paleolatitudinal biodiversity gradient for the late Jurassic in which we use a bootstrap approach to propagate paleolatitude and age uncertainty into the result.

## 1. Introduction

The study of paleoclimate, paleoceanography, and paleobiology depends for an important part on the interpretation of rocks and fossils, or geochemical tracers therein. However, these rocks and fossils are generally displaced relative to the location at which they were deposited. As a result of plate tectonic motions, as well

**Data availability statement:** All data are held in public repositories: https://doi.org/10.5281/zenodo.18183857 and https://doi:10.6084/m9.figshare.31021144.

**Funding:** Aard- en Levenswetenschappen, Nederlandse Organisatie voor Wetenschappelijk Onderzoek (NWO) Vici grant 865.17.001 to DJJvH. Aard- en Levenswetenschappen, Nederlandse Organisatie voor Wetenschappelijk Onderzoek (NWO) Veni grant 212.247 to L.M.B. HORIZON EUROPE European Research Council (ERC) starting grant (MindTheGap, grant No. 101041077 to EJ. HORIZON EUROPE European Research Council ERC consolidator grants (MATRICs, grant No. 101167761) to Pietro Sternai, supporting BV. HORIZON EUROPE European Research Council ERC consolidator grants (DISPERSAL, grant No. 101043268) to Alexis Licht, supporting BV. The funders played no role in study design, data collection, or analysis, decision to publish, or preparation of the manuscript.

**Competing interests:** No authors have competing interests.

as episodes of wholesale rotations of the solid Earth (crust and mantle) relative to the spin axis, known as true polar wander [1], the distribution of oceans and continents relative to each other and relative to the Earth's spin axis continuously changed throughout geological time. Interpreters of paleoclimate, paleoceanography, and paleobiology need to take these changes into account, for which they rely on paleogeographic reconstructions [2–5]. Such reconstructions, however, are typically made by a different scientific community – those studying plate tectonics and geodynamics – and it has proven to be a challenge to optimize communication between communities to ensure that the latest state-of-the-art is available for multidisciplinary research.

An important quantitative parameter that is provided by paleogeographic reconstructions is paleolatitude. Latitude, relative to the Earth's spin axis, determines the angle of solar insolation and thus climate (bearing in mind that the Milankovitch cycles caused by obliquity and precession modify solar insolation on ~20–40 kyr timescales). To allow a user-friendly estimation of paleolatitude, the online paleolatitude calculator of Paleolatitude.org was developed about a decade ago [6], was shortly after updated to version 2.0 that included the Paleozoic [7], and has since become a widely used tool in the study of paleoclimate and paleobiology studies [8–12]. The paleogeographic model behind that calculator, as well as the functionality of the calculator has now undergone some significant improvements.

In this contribution, we describe the upgrade of Paleolatitude.org to version 3.0. First, we upgrade to a fully-global paleogeographic model (dubbed the Utrecht Paleogeography Model), updated to comply with data published since 2015 on marine magnetic anomalies that describe the motions of major tectonic plates, and we have converted all ages to the most recent geological timescale [13]. We also integrate detailed regional kinematic reconstructions of rock units that are found in deformed orogenic belts such as in the Mediterranean region, Iran, Himalaya and Tibet, SE Asia, the Caribbean region, and of continental fragments that make up present-day Mongolia, China, and Indochina. In addition, Paleolatitude 3.0 uses a novel global paleomagnetic reference frame for the last 320 Ma based on a global apparent polar wander path (gAPWP) that is based on an improved statistical analysis that stays closer to the original data and that significantly decreases paleogeographic uncertainty [14]. In this paper, we provide the first upgrade of the paleomagnetic database behind that gAPWP, upgrading from gAPWP23 to gAPWP25. We provide a brief synopsis of global paleogeography since late Carboniferous Pangea and provide the GPlates-based [15] global plate model files of the Utrecht Paleogeography Model. Additionally, we describe the improved functionalities of the Paleolatitude.org online tool, which include batch calculations for large datasets and the quantification of paleolatitudinal uncertainty for each sample. Finally, we compare the results of Paleolatitude.org with other widely used models and illustrate the use our updated tool with example applications on paleobiological datasets.

## 2. Methods and innovations in paleolatitude reconstruction

Paleogeographic models are developed from relative global plate tectonic reconstructions (e.g., ref [16]), placed in a frame that positions the plates, oceans, and continents

relative to a chosen, 'fixed' reference. There are two types of such reference frames: the ones where plates are positioned relative to the mantle (mantle reference frames [17–20]), or relative to the Earth's magnetic field that on geological times aligns with the spin axis (paleomagnetic reference frames [14,21–23]. Mantle reference frames only 'see' motion of plates relative to the mantle, but not the common rotation of mantle and plates together relative to the spin axis (i.e., true polar wander). However, true polar wander may significantly change the distribution of global geography relative to the equator and poles, and is must therefore be included in paleogeographic reconstructions. Therefore, it is of importance that paleoclimate, paleoenvironment, or paleobiology is studied in paleogeographic context placed in the paleomagnetic reference frame, for only that frame provides quantitative information of the paleolatitude that determines the angle of solar insolation [6].

The main tool to quantitatively estimate the paleolatitude of a rock is paleomagnetism: the study of the Earth's magnetic field stored in rocks. The Earth's magnetic field is represented by field lines that, in a normal (or: reversed) field, point vertically out of (or: into) the Earth on the south pole, are horizontal pointing northward (or: southward) on the equator, and point vertically into (or: out of) the Earth at the north pole. Measuring the magnetic field stored in rocks thus provides a direct measure of the absolute paleolatitude at which it formed (provided that known sources of bias, scatter, and error are considered and corrected for [24,25]). However, as paleomagnetic data are not available for rocks of every age and every location, paleogeographic models use global plate reconstructions of relative plate motions that are then placed in a global reference frame (Fig 1).

With major plates being essentially rigid (i.e., internally not significantly deforming), the relative positions of major plates that are separated by ocean basins may be reconstructed from marine magnetic anomalies and fracture zones on the ocean floor [26]. Such reconstructions have typical uncertainties in the order of tens of kilometers [27]. Plates separated by ocean basins collectively form a global plate circuit [16,26]. This way, paleomagnetic data from one plate also constrain the position of all other plates in the plate circuit. All paleomagnetic data from plates connected in a global plate model may thus be used to collectively determine the paleoposition of the whole plate model relative to the spin axis, forming a global paleomagnetic reference frame [14,20–23,28]. The Paleolatitude.org tool used such plate models placed in a global paleomagnetic reference frame to predict the paleolatitude of any coordinate within the plate circuit, through time [6].

The original paleolatitude calculator Paleolatitude.org 1.0 [6] included three global paleomagnetic reference frames, each with the plate model that was used to compute the reference frame: the frame of Besse and Courtillot [22] for 0–200 Ma, the frame of Kent and Irving [23] for 50–220 Ma, and the frame of Torsvik et al. [21] for 0–320 Ma. In a subsequent upgrade to Paleolatitude.org 2.0, detailed in an online comment on the original paper [7], the calculator was extended to the early Paleozoic for the major continents, based on the spline-fitted apparent polar wander paths for pre-Pangean continents computed in Torsvik et al. [21].

In addition to plate motions of the rigid major plates, the geological record contains widespread evidence for distributed deformation. Such deformation includes extension (rifting), and shortening (orogenesis). Rock units of such deformed belts, including all mountain ranges of the Alpine-Himalayan belt and subduction-related fold-thrust belts of the circum-Pacific region were not yet included in the calculator. The underlying reason was that such reconstructions have additional and poorly quantifiable uncertainties. The paleoclimatic community, which was the target audience of the original calculator, tends to concentrate on rock records from stable plate interiors instead, e.g., from deep marine or passive margin shelf drill cores. However, those orogenic belts, which typically consist of deformed, sedimentary rocks offscraped from subducted oceanic plates or passive continental margins, provide far better outcrop access to geological records than most stable plate interiors do and these belts thus provide a rich paleontological record. In the last 15 years, detailed kinematic reconstructions of the orogenic belts have become available, especially because of the availability of GPlates open access plate reconstruction software [15]. This makes it now possible to upgrade Paleolatitude.org to include detailed reconstructions of orogenic belts.

In addition to developments in plate reconstructions, a new paleomagnetic reference frame for the last 320 Ma was developed [14]. The three frames that were included in the previous version of Paleolatitude.org all used the same

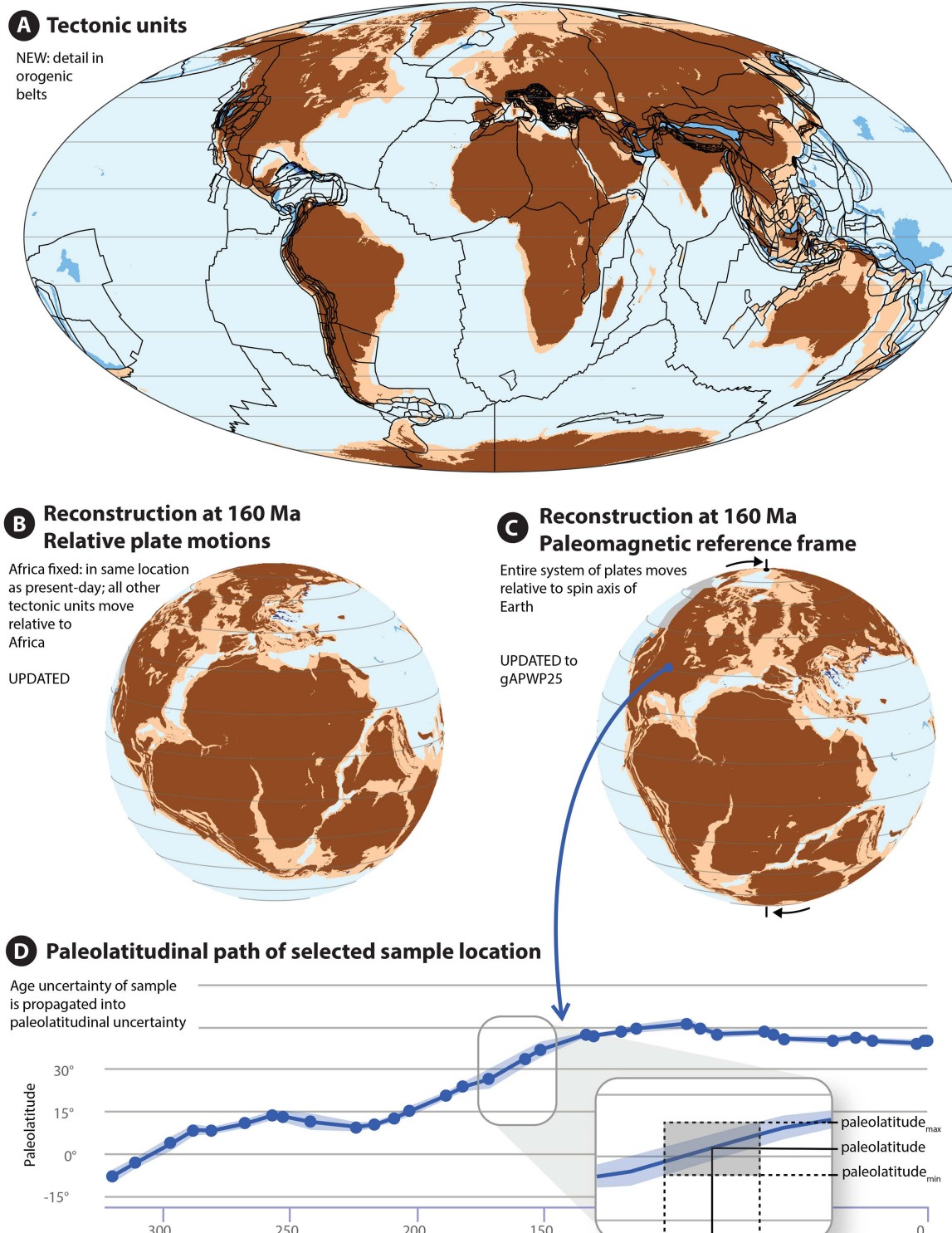

**Fig 1. Plate and paleogeographic reconstruction approach underlying the Paleolatitude.org calculator.**

underlying statistical approach, in which a series of paleopoles determined from stable plate interiors were averaged to form a global reference frame. The differences between the three frames thus stem from subtle differences in the underlying plate circuit reconstruction, and in the compilation of paleomagnetic poles used to compute the reference frame. However, the classical approach in paleomagnetism to combine paleomagnetic data into study-level poles, which contain an arbitrary number of data points, led to several flaws in the paleomagnetic quantification of relative tectonic motions [29], including irreproducibility and inflation of uncertainty [30]. Removing the arbitrary level of paleomagnetic poles and computing the global paleomagnetic reference frame at the site-level instead [30] gives equal weight to each individual measurement of the past magnetic field and led to gAPWP23 [14], which has much smaller uncertainty and higher reproducibility. Paleolatitude.org 3.0 thus uses this paleomagnetic reference frame as default. In addition, we here provide the first post-publication upgrade of the underlying dataset of this reference frame as explained in section 4.

## 3. Plate and orogen reconstruction approach

### 3.1. Plate circuit and intraplate deformation

The new default reconstruction in Paleolatitude.org 3.0 uses a global plate model that is based on marine magnetic anomalies and fracture zones of the modern ocean floor. The plate model underlying Paleolatitude 3.0 is the same as computed for the gAPWP23 paleomagnetic reference frame [14], updated with recently improved rotation poles for the Nazca relative to the Pacific Plate [31]. This plate circuit differs in details from the one underlying the APWPs used in the Paleolatitude 1.0 and 2.0: it includes more detailed reconstructions of ocean basins provided by the marine geophysical community in the last decade [32–35], and the age of all anomalies follows the latest geological timescale [13]. We refer the reader to the publication detailing gAPWP23 [14] for further details on the global plate model.

Besides regional reconstructions of orogenic belts, there is an increasing number of detailed reconstructions available of intra-plate rifting, i.e., the process of continental extension that precedes oceanic spreading and which may develop small microcontinental blocks adjacent to major continents [36–38]. Our global reconstruction incorporates reconstructions of microcontinental blocks but has not incorporated all details of reconstructions of rifting from passive margins yet. For some margins, such reconstructions are available [36,39] and they may be incorporated in future updates. Typical pre-break-up extension amounts ~150 km per passive margin [20], meaning that pre-extensional paleolatitudes of rocks on distal passive margins in our reconstructions may be up to 2° off if rifting had a N-S component.

Orogenic belts are regions where crust and lithosphere deformed, and where the hypothesis of plate rigidity that underlies plate reconstructions fails. Those belts contain rock units that have moved relative to their stable neighboring plates, and a much larger level of detail is needed to restore paleolatitude. Orogenic belts are in this reconstruction categorized in two classes: 'intraplate' orogens, that result from shortening of full lithospheric sections, and 'accretionary' orogens that consist of rock units offscraped from lithosphere that disappeared into the mantle by subduction [40].

Intraplate orogens, such the Andes, the Rocky Mountains, the Tibetan Plateau, the Tian Shan, or the Atlas Mountains, are regions where lithosphere was compressed, and crust was thickened and uplifted. The amount of deformation is typically limited – the largest intraplate shortening was restored in the Tibetan Plateau, and amounts to ~1000 km [41], whereas the Andes only recorded up to ~400 km [42,43], and the Atlas only accommodated some tens of kilometers of shortening [44]. Intraplate deformation may also be extensional, for instance in back-arc basin settings, where previously thickened orogens may be stretched, such as in the Aegean region [45], the Basin and Range province [46], or the Sea of Japan [47]. Because back-arc basins often form from previously thickened orogenic crust, they may accommodate more continental extension before oceanic spreading than continents – for instance, the Aegean and Basin and Range regions already experienced ~400 km of extension and no oceanization has started yet [46,48].

Continental intraplate deformation may thus move rock units relative to stable plate interiors by a few degrees, and in extreme cases up to ~10°. We reconstruct such motions using a reconstruction protocol that uses structural geological, stratigraphic, and geochronological evidence to reconstruct (i) extensional deformation, which achieves its largest

geological record at the end of the deformation and is thus the most complete; (ii) displacement along strike-slip faults, whereby the motion direction is well-constrained but uncertainty may exist on the amount and timing of motion, and (iii) shortening deformation, whereby the least complete geological record is achieved at the end of deformation and only a minimum estimate of shortening may be made, of which the direction of shortening may also have uncertainty. Subsequently (iv) a reconstruction is made that is geometrically feasible without violating constraints, that is (v) tested against and if necessary iterated within the constraints of steps (i) to (iii) based on paleomagnetic data that demonstrate paleolatitudinal and vertical axis rotations [49,50].

Oceanic crust may also undergo deformation, especially in oceanic subduction-zone settings, but this deformation is typically more confined than in continental crust. Contractional deformation is typically restricted to weak crust of volcanic arcs in the upper plate of subduction zones [51]. Extensional deformation in upper oceanic lithosphere is common, e.g., in the Scotia Sea region [52,53], the Caribbean Plate [54] or in the Philippine Sea Plate [55] and often leads to formation of microplates separated by back-arc basin ridges. In addition, forearc slivers may form that displace upper plate oceanic or arc fragments relative to stable plate interiors [56]. Reconstruction of this type of deformation follows similar protocols as for deformed upper plate continental crust.

The preservation potential of upper plate oceanic lithosphere in the geological record is limited: eventually, it will subduct. An exception is oceanic lithosphere of the forearc, close to the plate contact. When accretionary prisms form below oceanic overriding plate lithosphere, or when continental margins arrive in a trench, the oceanic forearc may become uplifted and be protected from later subduction. Such uplifted, or 'obducted' oceanic forearcs then become preserved as ophiolites [57,58]. Those ophiolites are often associated with deep-marine, pelagic oozes that hold important geological records of past oceanography, and our reconstruction has included their pre-obduction plate motion history to unlock such potential for global oceanography and planktonic biogeography. Key examples of orogens rich in obducted ophiolites include the Balkans, Cyprus, Anatolia, and Oman [59–62], the Philippines [63], New Guinea and New Caledonia [64], Cuba [65], and California [66], among many other examples.

### 3.2. Accretionary orogenesis and subducted-plate reconstruction

Rock units in accretionary orogens may have travelled far larger distances relative to stable plate interiors. Accretionary orogens consist of rock units that were once part of now-subducted plates, and that were offscraped at subduction zones and accreted to upper plates, escaping subduction. Such accretionary orogens include the Pyrenees, Alps, Apennines, Dinarides, Hellenides, or Taurides in the Mediterranean region [67–70], the Zagros mountains in Iran [71], the Himalaya [72], large parts of SE Asia [73–75] much of Japan [76] and New Zealand [77], South Alaska [78] or California [66].

Rocks that accreted in such orogens may be derived from oceanic lithosphere, in which case they contain a history from the formation at a mid-oceanic ridge to the accretion into the orogen at a subduction zone [79]. Alternatively, they may be derived from continental lithosphere – typically passive continental margins or microcontinents. In that case, the accreted slices may contain a basement that underwent an earlier orogenic history, and a sequence that represents continental rifting, a passive margin evolution, and the accretion to an orogen when the continental margin went down into a trench [40]. Reconstructions of accretionary orogens first restore post-accretion intra-plate deformation [48], and subsequently reconstruct accreted rock units as part of the original downgoing plate prior to the moment of accretion. The maximum age of accretion is provided by the deposition of a coarsening-upward clastic sedimentary series derived from the upper plate (flysch, molasse) in the top of the stratigraphic sequence of the accreted unit, which marks the arrival of the units in a foreland basin/trench. The minimum age is determined by the oldest metamorphism, magmatism, contractional deformation, or upper plate sedimentation (i.e., forearc basin sedimentation [80]) that affected the unit and that indicates that it had become part of the upper plate [40,81]. This time interval is typically constrained within a few million years and depending on the rate of convergence between the adjacent plates, may add a few degrees of uncertainty to the reconstructed position of an accreted unit on the original downgoing plate.

Importantly, none of the reconstruction protocols included any paleoclimatic, paleoenvironmental, or paleobiological interpretations. The reconstructions selected for Paleolatitude.org 3.0 provide independent paleogeographic input for such studies without introducing circular reasoning.

All kinematic reconstructions were made in GPlates plate reconstruction software [15] and the global plate reconstruction underpinning the Paleolatitude.org 3.0 model is provided in the Supplementary information in S1 File (available in a DOI). In basin and orogen reconstructions, area change occurs due to deformation. For the paleolatitude calculator we divide such deforming regions in rigid polygons that may partly overlap (when extension is reconstructed) or be separated by gaps (when shortening is reconstructed) when reconstructed backwards in time. This division into polygons is an obvious, but practical, simplification and adds to the uncertainty of the region. However, for intensely deformed regions, we used polygons on the scale of tens of kilometers (yielding thousands of polygons in the model) and overlaps rarely exceed 100 km (i.e., max ~ 1° in paleolatitude) (Fig 1). Polygons in orogens encompass rock units with a common paleogeographic origin, which are bounded by faults. Polygons are typically pre-orogenic stratigraphic sequences that were incorporated in orogens as major thrust slices, called nappes. These may have become deformed, metamorphosed, intruded by magmatic rocks, and subsequently overlain by sedimentary basins. Each polygon in an orogen is named after the nappe or tectonic block it represents. Younger magmatic rocks or sedimentary basins are reconstructed with the nappes they intrude or overlie and are not marked as separate polygons.

Despite the detail in the reconstructions of orogenic belts, simplifying their geological complexity is inevitable. The true spatial distribution and geological structure of rock units, as seen on a geological map cannot be fully captured into a global 2D model. In most cases, this will not significantly affect the estimation of the paleolatitude of a specific site. We acknowledge, however, that due to small georeferencing errors, we may have misplaced tectonic boundaries by a few kilometers, which would cause a coordinate to fall in a wrong polygon, in which case an incorrect paleolatitude may be provided. Similarly, inaccuracies in sampling locations of a fossil or rock from an orogenic belt may place them in an incorrect tectonic unit. Such potential errors are typically not more than a few degrees but need to be considered when using the reconstruction.

The reconstruction uses a series of regional tectonic reconstructions of intraplate deformation, back-arc basin development, and accretionary orogenesis, including of the Scotia Sea [82]; the Andes mountains [43], the Caribbean region [49,83], the western United States [46,84], the Mediterranean region [48,50,85,86], the Central Tethysides of the Iran-Afghanistan [87,88], Oman [89], the Tibetan Plateau and Himalaya [41,90,91], SE Asia [74], the Junction region of the Pacific and Tethys realms around the Philippine Sea Plate and the SW Pacific back-arc basins [92,93], the NW Pacific and Bering Sea region [94,95], and the China Blocks and the Tibetan and Sundaland terranes [96–100]. A detailed reconstruction of the Canadian Cordillera, Alaska, and pre-late Cretaceous NE Siberia (Kolyma-Omolon) is not yet included because none is available that follows our reconstruction protocol – such a reconstruction will be incorporated in a next upgrade of the Paleolatitude.org tool. In the Supplementary information in S1 File of this paper (available in a DOI), we provide GPlates files of the rigid polygon model that underpins the Paleolatitude.org tool, and a paleogeography version that shows the paleogeographic distribution of oceanic and continental lithosphere of the Utrecht Paleogeography Model.

## 4. gAPWP25: Updated paleomagnetic reference frame

Here, we provide the first update of the site-based global apparent polar wander path (gAPWP) of Vaes et al. [14] for the past 320 Myr. This updated path, named *gAPWP25*, serves as the new default paleomagnetic reference frame of Paleolatitude.org 3.0, and related online tools including Paleomagnetism.org [101,102] and APWP-online.org [103].

We made the following modifications to the paleomagnetic database that underlies the site-level based gAPWP. First, we corrected typographical errors in entry names, sampling locations, and other parameters. Second, we revised the ages of four North American datasets according to constraints pointed out in recent compilations [104,105]. All modifications to the database are documented in the change log. We further compiled all paleomagnetic poles published since 2022 that

were obtained from rocks younger than 320 Ma exposed in stable continental interiors. From this compilation, we added 26 datasets that satisfy the selection criteria of Vaes et al. [14] to the global data compilation (Table 1). Sediment-derived datasets were accepted if they either meet the reliability criteria for inclination shallowing-corrected poles [106], receiving a reliability grade 'A' or 'B', or pass both the bootstrap reversal test of Heslop et al. [107] and the SVEI test of Tauxe et al. [108]. In addition, six datasets published prior to 2000 that satisfy the selection criteria were added, two of which were previously excluded in gAPWP23 [14] but are included following a positive SVEI and reversal test result. In total, 32 entries were added (~10%) to the database used to compute the updated global APWP. The complete database is provided in the Supplementary Files (available in a DOI), and on APWP-Online.org, where also future further updates will also be logged.

The updated global APWP was computed using the approach described in Vaes et al. [14] and is provided in coordinates of South Africa in Table 2 (for versions in the coordinates of other major continents, see the Supplementary Files (available in a DOI)). The gAPWP25 shows only minor differences with its predecessor gAPWP23 (Fig 2). The largest angular differences (~1.5°-2.5°) are observed for three time intervals: the Late Cretaceous, latest Jurassic and Early Triassic (Fig 2). These intervals are characterized by relatively low data density, which increases the influence of newly added datasets. Nevertheless, all reference poles of gAPWP25 have overlapping 95% confidence regions with those of gAPWP23. Likewise, estimated APW rates show no significant changes. Absolute plate motions in the paleomagnetic reference frame, and estimated changes in paleolatitude over time, will therefore remain very similar to those predicted by gAPWP23, albeit with slightly smaller uncertainties due to the increased amount of data. The paleomagnetic reference frame can be used for any global or regional plate reconstruction in the GPlates software [15] by adding total reconstruction poles provided in Table 3 to the rotation file. A ready-to-use rotation file is included in the Supplementary Files (available in a DOI), which ties South Africa (plate ID 701) to the spin axis (plate ID 001) for the past 320 Myr.

## 5. A brief synopsis of global paleogeography since the Carboniferous

As with any global paleogeographic model, it is inevitable that our version of it made choices that are under debate and scrutiny. Below, we describe the general characteristics of the model, and the sources that were used for the compilation. All GPlates reconstruction files are available in the Supplementary information in S1 File (available in a DOI), openly available for anyone to modify.

Earth's changing paleogeography may at first order be described in the terminology used for supercontinents: a dispersing set of continents that enclose an internal ocean (the Tethys), and that consume an external ocean (the Panthalassa or Paleo-Pacific) [137,138]. Most of the modern continental crust, except Siberia (until ~250 Ma ago) and the China Blocks (until ~140 Ma ago) [99,100] was joined together in the Late Carboniferous and Permian in the supercontinent Pangea. This is the oldest part of the global reconstruction covered by our paleolatitude calculator. From the Jurassic onwards, the continents dispersed by the opening of the Atlantic Ocean and associated proto-Caribbean and Alpine Tethys oceans, as well as the Indian and Southern Oceans (Fig 3). The opening of the Atlantic Ocean and western Southern Ocean occurred at the expense of the external, Panthalassa Ocean. Lithosphere of this ocean basin was consumed at circum-Panthalassa subduction zones and remains of these plates are now found in the circum-Pacific accretionary orogens. The opening of the Indian and eastern Southern Oceans occurred at the expense of the internal, Tethys Ocean, which closed and formed the Alpine-Himalayan-Indonesian accretionary orogen (Fig 3).

### 5.1. Exterior Panthalassa ocean and its margins

The exterior, Panthalassa Ocean consisted mostly of oceanic plates that spread relative to each other and were consumed at subduction zones, both intra-oceanic [139–142], as well as along the margins of the Pangea continents North and South America, Antarctica, Australia as well as Siberia and the China Blocks. Back in time, the modern plates underlying the Pacific Ocean covered an increasingly smaller area. The remaining area was mostly occupied by oceanic lithosphere that has since been lost to subduction. Geological remains of these 'lost' Panthalassa plates consist of remnants of formerly intraoceanic

**Table 1. List of data that were added up of Vaes et al. [14] to upgrade to gAPWP25. For total database, see Supplementary information in S1 File** (available in a DOI), or apwp-online.org. $Age_{min}$ and $Age_{max}$ = lower and upper boundaries of age uncertainty range; Slat/Slon = latitude and longitude of (mean) sampling location; N = number of paleomagnetic sites used to compute the paleopole; A95 = radius of the 95% confidence circle about the mean of the distribution of VGPs; K = Fisher [109] precision parameter of the distribution of VGPs; Plat/Plon = paleopole latitude and longitude (south pole); Rlat/Rlon = paleopole latitude and longitude in coordinate frame of South Africa; f = flattening factor (only for sedimentary data), pstd = standard deviation of the assumed normal distributed co-latitudes, obtained from E/I correction [110] (only for sedimentary data); Key to references: a = [111], b = [112], c = [113], d = [114], e = [115], f = [116], g = [117], h = [118], i = [119], j = [120], k = [121]; l = [122], m = [123], n = [124]; o = [125]; p = [126]; q = [127], r = [128], s = [129], t = [130], u = [131], v = [132], w = [133], x = [134], y = [135], z = [136].

| Name | Agemin | Agemax | age | Slat | Slon | N | K | A95 | plat | plon | Rlat | Rlon | lithology | f | p_std | reference |
|---|---|---|---|---|---|---|---|---|---|---|---|---|---|---|---|---|
| Mt Ruapehu volcano, Aotearoa New Zealand | 0 | 0.01 | **0.005** | −39.3 | 175.6 | 18 | 59.1 | 4.5 | −85.1 | 77.5 | −85.1 | 77.5 | igneous | | | a |
| Tres Vírgenes Volcanic Complex, Baja California, Mexico | 0.02 | 0.30 | **0.2** | 27.5 | −112.6 | 12 | 12.0 | 13.0 | −80.9 | 333.0 | −80.9 | 333.1 | igneous | | | b |
| Trindade Island, offshore Brazil | 0.06 | 0.8 | **0.4** | −20.5 | −29.3 | 14 | 12.2 | 11.9 | −79.1 | 267.4 | −79.1 | 267.7 | igneous | | | c |
| Andacollo volcanics, Argentina | 0.9 | 3.8 | **2.4** | −37.2 | −70.8 | 17 | 30.2 | 6.6 | −84.9 | 31.5 | −84.5 | 33.2 | igneous | | | d |
| Caviahue-Copahue Volcanic Complex, Northern Patagonia | 0.0 | 5.6 | **2.8** | −37.9 | −71.0 | 42 | 26.4 | 4.4 | −84.3 | 251.4 | −84.7 | 253.6 | igneous | | | e |
| Eyjafjarðardalur basalts, Iceland | 2.6 | 8.0 | **5.3** | 65.5 | −18.8 | 114 | 11.9 | 4.0 | −82.0 | 5.0 | −82.1 | 7.4 | igneous | | | f |
| Vogelsberg volcanics, Germany | 15.2 | 17.6 | **16.4** | 50.5 | 9.2 | 116 | 18.2 | 3.2 | −84.5 | 341.9 | −84.2 | 359.8 | igneous | | | g |
| Imnaha and Grande Ronde basalts, US | 16.0 | 17.0 | **16.5** | 45.8 | −116.8 | 30 | 15.0 | 6.8 | −85.0 | 335.0 | −85.7 | 343.7 | igneous | | | h |
| Sleat Peninsula dykes, Isle of Skye, UK | 53.9 | 61.7 | **57.8** | 57.00 | −5.90 | 24 | 21.1 | 6.6 | −75.2 | 1.8 | −69.4 | 32.0 | igneous | | | i |
| South Rewa Basin dykes, India | 64.0 | 67.0 | **65.5** | 23.8 | 81.7 | 13 | 33.9 | 6.9 | −42.0 | 109.3 | −77.3 | 66.3 | igneous | | | j |
| Uberaba Formation, Brazil | 72.2 | 76.0 | **74.1** | −19.8 | −47.9 | 120 | 13.4 | 3.7 | −85.2 | 336.0 | −75.1 | 51.3 | sedimentary | 0.6 | 3.2 | k |
| Alkaline dykes, Santos-Rio de Janeiro coast, Brazil | 80.0 | 88.0 | **84.0** | −23.9 | −45.4 | 44 | 44.0 | 3.0 | −81.2 | 319.7 | −70.8 | 43.2 | igneous | | | l |
| Okhotsk-Chukotka Volcanic Belt, Siberia | 83.7 | 88.6 | **86.2** | 66.9 | 170.0 | 57 | 14.1 | 5.2 | −76.8 | 350.0 | −67.6 | 45.5 | igneous | | | m |
| Granite Mountain, Arkansas, US | 87.5 | 91.5 | **89.5** | 34.7 | 267.7 | 5 | 21.0 | 17.1 | −77.8 | 351.4 | −70.5 | 49.6 | igneous | | | n |
| Ramon volcanics, Israel | 112.6 | 119.1 | **115.8** | 30.5 | 34.7 | 46 | 35.6 | 3.6 | −57.2 | 72.3 | −57.5 | 72.4 | igneous | | | o |
| Parana basalts – Gramado & Herveiras regions, Brazil | 133.6 | 135.0 | **134.3** | −29.4 | −52.6 | 37 | 56.6 | 3.2 | −82.8 | 45.2 | −48.3 | 81.1 | igneous | | | p |
| Puerta Curaco section, Tithonian, Neuquen, Argentina | 143.1 | 149.2 | **146.2** | −37.4 | 290.1 | 27 | 50.4 | 4.0 | −81.1 | 108.6 | −49.0 | 91.7 | sedimentary | 1.00 | 0.00 | q |
| Notre Dame Bay dikes 2 | 146.1 | 150.0 | **148.1** | 49.5 | 304.9 | 15 | | | −73.9 | 21.0 | −48.8 | 94.3 | igneous | | | r |
| Penatecaua Formation, Brazil | 200.4 | 202.4 | **201.4** | −3.0 | −54.0 | 30 | 48.0 | 3.8 | −77.5 | 260.1 | −63.1 | 59.7 | igneous | | | s |
| Mercia Mudstone Group (Haven Cliff), England, UK | 205.0 | 212.0 | **208.5** | 50.7 | −3.2 | 74 | 24.0 | 3.4 | −54.8 | 287.7 | −66.0 | 53.2 | sedimentary | 0.65 | 3.56 | t |
| Mercia Mudstone Group (ML, SH, MB, SE), England, UK | 212.0 | 224.0 | **218.0** | 50.7 | −3.2 | 83 | 21.5 | 3.4 | −56.2 | 295.4 | −63.6 | 55.5 | sedimentary | 0.50 | 4.07 | t |

*(Continued)*

**Table 1.** (Continued)

| Name | Agemin | Agemax | age | Slat | Slon | N | K | A95 | plat | plon | Rlat | Rlon | lithology | f | p_std | reference |
|---|---|---|---|---|---|---|---|---|---|---|---|---|---|---|---|---|
| Mercia Mudstone Group (MS, MD, MW), England, UK | 227.3 | 240.4 | **233.9** | 50.7 | −3.2 | **70** | 29.0 | 3.2 | **−51.4** | **308.2** | **−55.5** | **49.5** | sedimentary | 0.85 | 2.21 | t |
| Otter Sandstone Fm, Devon, England, UK | 239.5 | 244.2 | **241.9** | 50.6 | −3.3 | **31** | 20.1 | 5.9 | **−55.2** | **326.0** | **−47.3** | **61.8** | sedimentary | 1.00 | 0.00 | u |
| Musschelkalk carbonates, Poland | 237.0 | 246.7 | **241.9** | 50.0 | 19.5 | **28** | 65.4 | 3.4 | **−51.0** | **323.0** | **−46.9** | **55.1** | sedimentary | 1.00 | 0.00 | v |
| Abinskaya Group, Siberian large igneous province | 250.8 | 252.2 | **251.5** | 54.3 | 84.1 | **33** | 20.2 | 5.7 | **−59.0** | **340.3** | **−42.3** | **71.6** | igneous | | | w |
| Nzalet el Lararcha, Morocco | 276.5 | 277.7 | **277.1** | 32.3 | 352.4 | **12** | 30.4 | 8.0 | **−49.8** | **45.3** | **−49.6** | **47.4** | igneous | | | x |
| Mechraa Ben Abbou, Morocco | 284.0 | 292.8 | **288.4** | 32.7 | 352.2 | **15** | 45.8 | 5.7 | **−45.1** | **41.5** | **−47.4** | **47.3** | igneous | | | x |
| Kenifra, Morocco | 283.4 | 295.7 | **289.6** | 33.0 | 354.3 | **12** | 33.4 | 7.6 | **−34.4** | **59.4** | **−45.0** | **43.4** | igneous | | | x |
| Tiddas, Morocco | 281.7 | 295.1 | **288.4** | 33.6 | 353.8 | **14** | 42.9 | 6.1 | **−47.6** | **45.3** | **−33.9** | **60.9** | igneous | | | x |
| Chougrane, Morocco | 292.1 | 311.8 | **302.0** | 33.0 | 353.7 | **12** | 54.8 | 5.9 | **−37.6** | **63.8** | **−37.0** | **65.4** | igneous | | | x |
| *From Vaes et al. (2023) database* | | | | | | | | | | | | | | | | |
| Monteregian Hills intrusives | 122.8 | 126.1 | **124.5** | 45.3 | 286.8 | **70** | 29.0 | 3.2 | **−72.4** | **11.0** | **−52.1** | **81.1** | igneous | | | y |
| Heming limestone, France | 237.0 | 246.7 | **241.9** | 48.7 | 7.0 | **58** | | | **−54.3** | **320.6** | **−49.7** | **58.6** | sedimentary | 1.00 | 0.00 | z |

subduction zones and fragments of plates that were trapped between or adjacent to continents, rock units that broke off circum-Panthalassa continents and were later re-accreted, and accretionary prisms offscraped of Panthalassa lithosphere. Examples of the latter are the earlier mentioned records of accretion in the orogens of New Zealand, Japan, Alaska, and California. These records are often sparse or so narrow that we have not reconstructed these in detail yet.

Examples of trapped oceanic lithosphere include the modern Caribbean Plate and circum-Caribbean accreted arc fragments now exposed in, e.g., Colombia, Venezuela, Cuba, and Nicaragua, which are fragments of the Jurassic Farallon plate [49,143–145]. The Aleutian Basin in the Bering Sea region likely contains Cretaceous back-arc basin crust that formed above an intra-oceanic subduction zone whose arc remains are now found on Kamchatka, the circum-Sea of Okhotsk region, and northern Japan [94]. This piece of oceanic lithosphere likely became trapped between Alaska and Siberia upon initiation of the Aleutian subduction zone at ~55−50 Ma (Fig 3).

Arc or continental fragments that became separated from the Panthalassa margins through formation of back-arc basins are now prominent in the SW Pacific region, where the continent of Zealandia broke off Antarctica and Australia, and back-arc basins formed between Zealandia and the Pacific realm [93,146,147] (Fig 3). Similar systems (de)formed and displaced the circum-Philippine Sea plate records as well as ophiolite complexes of New Guinea and the Philippines [92,148,149]. The Cordilleran orogen of western North America likely underwent similar processes but its geological history remains debated. Mexico hosts the remains of the Guerrero Arc that became separated and reconnected with North America through the opening and closures of the Late Jurassic to Cretaceous Arperos back-arc basin [150,151]. Farther north in western Canada and Alaska are records that also indicate systems like this, but as explained before, we have not incorporated this region yet in our reconstruction and refer the reader to a selection of publications [142,152–157] that cover some of the many different views on NE Panthalassa/Cordilleran history. The last region with widespread orogenic deformation in the circum-Pacific region is the Scotia Sea region. This deformed belt formed by westward subduction of

**Table 2. Global apparent polar wander path of Vaes et al. [14] upgraded to gAPWP25 using the additional data shown in Table 1, calculated using a 20 Ma sliding window. For each window, the mean age of the re-sampled VGPs in that window is provided. N and P95 are the average number of re-sampled VGPs that fall within the time window and the 95% confidence region of the reference pole (in degrees). Mean K, CSD and E are the average [109] precision parameter, circular standard deviation, and elongation of the re-sampled VGPs, respectively.**

| Window | Age | N | P95 | Longitude | Latitude | Mean K | Mean CSD | Mean E |
|---|---|---|---|---|---|---|---|---|
| 0 | 1.4 | 1960.2 | 0.7 | 324.3 | −89.3 | 18.8 | 18.7 | 1.06 |
| 10 | 4.6 | 2915.0 | 1.2 | 346.8 | −87.8 | 17.8 | 19.2 | 1.08 |
| 20 | 21.5 | 1261.9 | 1.1 | 12.5 | −82.7 | 17.3 | 19.5 | 1.08 |
| 30 | 28.2 | 1087.8 | 1.0 | 23.3 | −80.8 | 18.0 | 19.1 | 1.08 |
| 40 | 37.5 | 475.2 | 1.4 | 26.0 | −79.6 | 19.6 | 18.3 | 1.14 |
| 50 | 56.1 | 1119.3 | 1.0 | 31.1 | −75.1 | 16.3 | 20.1 | 1.11 |
| 60 | 60.1 | 1744.2 | 0.8 | 35.3 | −73.7 | 16.5 | 20.0 | 1.08 |
| 70 | 65.7 | 1029.5 | 1.3 | 40.5 | −73.4 | 16.9 | 19.7 | 1.11 |
| 80 | 81.0 | 574.3 | 1.8 | 49.6 | −72.5 | 21.8 | 17.4 | 1.13 |
| 90 | 88.8 | 524.9 | 1.3 | 60.7 | −68.7 | 23.2 | 16.8 | 1.16 |
| 100 | 94.3 | 214.1 | 2.4 | 71.6 | −64.3 | 22.6 | 17.1 | 1.22 |
| 110 | 115.1 | 300.3 | 1.4 | 79.5 | −57.8 | 30.1 | 14.8 | 1.23 |
| 120 | 120.3 | 568.2 | 1.1 | 79.4 | −55.0 | 29.2 | 15.0 | 1.15 |
| 130 | 130.7 | 895.1 | 0.8 | 82.4 | −50.8 | 33.5 | 14.0 | 1.09 |
| 140 | 135.2 | 706.6 | 0.9 | 84.6 | −49.8 | 35.7 | 13.6 | 1.12 |
| 150 | 151.0 | 196.4 | 2.2 | 86.6 | −51.9 | 20.5 | 17.9 | 1.23 |
| 160 | 158.7 | 149.0 | 3.1 | 83.2 | −55.8 | 14.7 | 21.2 | 1.23 |
| 170 | 172.6 | 112.0 | 3.6 | 78.8 | −59.0 | 14.7 | 21.2 | 1.29 |
| 180 | 182.1 | 319.3 | 1.7 | 79.5 | −64.2 | 19.0 | 18.6 | 1.29 |
| 190 | 189.7 | 470.1 | 1.5 | 75.3 | −65.8 | 18.9 | 18.7 | 1.18 |
| 200 | 203.8 | 1482.2 | 1.7 | 62.9 | −65.3 | 12.7 | 22.7 | 1.09 |
| 210 | 209.7 | 2446.5 | 1.3 | 58.0 | −63.0 | 15.2 | 20.8 | 1.07 |
| 220 | 217.5 | 1700.9 | 1.1 | 53.9 | −59.8 | 22.1 | 17.2 | 1.07 |
| 230 | 225.9 | 670.2 | 1.5 | 53.1 | −56.5 | 23.3 | 16.8 | 1.11 |
| 240 | 241.3 | 387.9 | 1.9 | 57.4 | −48.6 | 18.5 | 18.8 | 1.18 |
| 250 | 252.4 | 1139.1 | 1.9 | 61.8 | −43.2 | 14.2 | 21.5 | 1.10 |
| 260 | 257.0 | 1233.0 | 1.7 | 62.6 | −42.1 | 15.6 | 20.5 | 1.10 |
| 270 | 268.7 | 645.7 | 1.7 | 58.4 | −41.1 | 26.4 | 15.8 | 1.15 |
| 280 | 281.4 | 844.6 | 1.3 | 56.8 | −37.8 | 29.6 | 14.9 | 1.24 |
| 290 | 288.4 | 795.1 | 1.9 | 57.7 | −35.5 | 29.7 | 14.9 | 1.15 |
| 300 | 297.8 | 427.4 | 2.5 | 52.4 | −31.3 | 24.8 | 16.3 | 1.27 |
| 310 | 311.0 | 375.7 | 2.6 | 45.7 | −26.8 | 18.4 | 18.9 | 1.40 |
| 320 | 320.2 | 428.5 | 2.7 | 39.9 | −28.3 | 13.5 | 22.0 | 1.34 |

Atlantic Southern Ocean lithosphere below South America and eastward subduction of Pacific Southern ocean lithosphere below the Antarctic peninsula, rifting fragments off both continental overriding plates and dispersing these via opening of multiple small, oceanic back-arc basins since the Eocene [52,53,82,158].

## 5.2. Interior Tethys oceans, microcontinents, and margins

The paleogeography of the interior ocean of the Tethyan realm is in general more complex than that of the exterior ocean, due to the repeated rifting of continental fragments off one margin – often the southern – and their accretion to the other

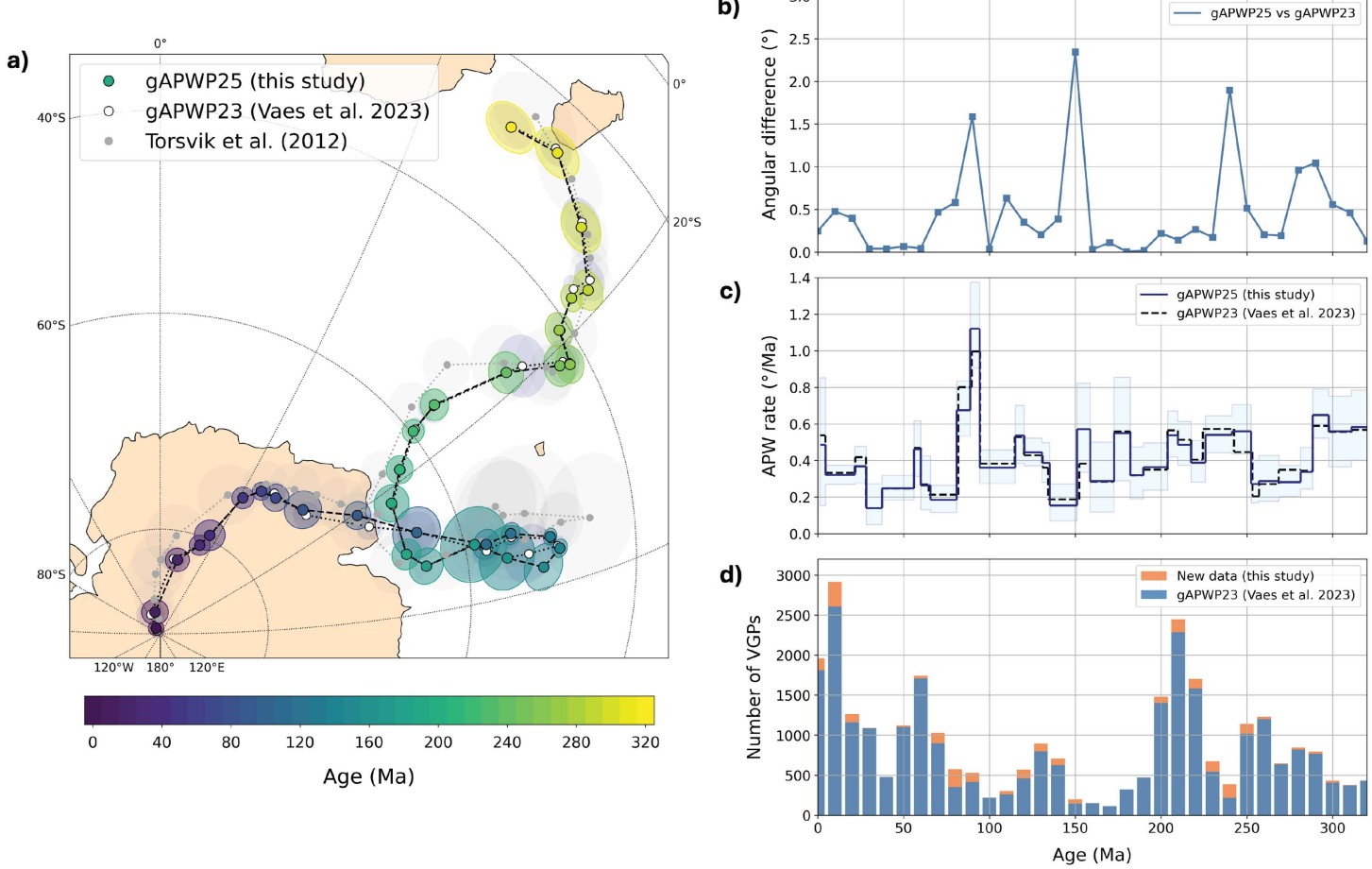

**Fig 2. Global apparent polar wander path of Vaes et al. [14] upgraded to gAPWP25, using the additional data listed in Table 1.** For the new APWP and the associated paleomagnetic reference poles, see Tables 2 and 3, respectively.

– northern – margin [159–161]. Particularly in the Permian and Triassic, the Pangea-Tethys system was essentially in a mode of self-subduction [162], whereby the consumption of oceanic lithosphere below one margin led to the break-up of the opposite margin, forming a new ocean basin whose growth was accommodated by subduction of the older ocean [162]. When Pangea started breaking up in the Jurassic, this mode of rifting on one side and collision on the other side continued, such that continental fragments migrated northward from the southern to the northern hemisphere over distances that increased towards the east [74,91,163]. Many of the orogens of the Alpine-Himalayan mountain belt contain remnants of such microcontinental fragments. The Alpine-Himalayan orogen is divided into E-W trending segments separated by ancient transform fault systems. These segments reflect the opening of ocean basins at different times. This is somewhat analogous to the modern Atlantic Ocean that is compartmentalized into four segments: the South Atlantic, formed by Cretaceous separation of Africa and South America; the Central Atlantic, formed in Jurassic time by separation of Africa from North America; the 'Iberian' Atlantic and Bay of Biscay, formed by Jurassic separation of Iberia from New-foundland and Eurasia; and north Atlantic, formed by Cenozoic separation of Eurasia from North America/Greenland [16]. In the Neotethyan realm, the different segments of the plate boundary system coincide with the Mediterranean, Iranian, Tibetan, and SE Asian regions. The former two formed by oceans opening and closing between Gondwana and Eurasia.

**Table 3. Paleomagnetic reference frame based on the updated gAPWP25 [14], rotating South Africa (701) into the coordinates of the Earth's spin axis (001). See Supplementary information in S1 File (available in a DOI) for a version in GPlates.rot file format.**

| PlateID | Age | Euler_lat | Euler_lon | Euler_ang | Fixed plateID |
|---|---|---|---|---|---|
| 701 | 0 | 0 | 90 | 0 | 1 |
| 701 | 1.4 | 0 | 54.3 | 0.7 | 1 |
| 701 | 4.6 | 0 | 76.8 | 2.2 | 1 |
| 701 | 21.5 | 0 | 102.5 | 7.3 | 1 |
| 701 | 28.2 | 0 | 113.3 | 9.2 | 1 |
| 701 | 37.5 | 0 | 116.0 | 10.4 | 1 |
| 701 | 56.1 | 0 | 121.1 | 14.9 | 1 |
| 701 | 60.1 | 0 | 125.3 | 16.3 | 1 |
| 701 | 65.7 | 0 | 130.5 | 16.6 | 1 |
| 701 | 81.0 | 0 | 139.6 | 17.5 | 1 |
| 701 | 88.8 | 0 | 150.7 | 21.3 | 1 |
| 701 | 94.3 | 0 | 161.6 | 25.7 | 1 |
| 701 | 115.1 | 0 | 169.5 | 32.2 | 1 |
| 701 | 120.3 | 0 | 169.4 | 35.0 | 1 |
| 701 | 130.7 | 0 | 172.4 | 39.2 | 1 |
| 701 | 135.2 | 0 | 174.6 | 40.2 | 1 |
| 701 | 151.0 | 0 | 176.6 | 38.1 | 1 |
| 701 | 158.7 | 0 | 173.2 | 34.2 | 1 |
| 701 | 172.6 | 0 | 168.8 | 31.0 | 1 |
| 701 | 182.1 | 0 | 169.5 | 25.8 | 1 |
| 701 | 189.7 | 0 | 165.3 | 24.2 | 1 |
| 701 | 203.8 | 0 | 152.9 | 24.7 | 1 |
| 701 | 209.7 | 0 | 148.0 | 27.0 | 1 |
| 701 | 217.5 | 0 | 143.9 | 30.2 | 1 |
| 701 | 225.9 | 0 | 143.1 | 33.5 | 1 |
| 701 | 241.3 | 0 | 147.4 | 41.4 | 1 |
| 701 | 252.4 | 0 | 151.8 | 46.8 | 1 |
| 701 | 257.0 | 0 | 152.6 | 47.9 | 1 |
| 701 | 268.7 | 0 | 148.4 | 48.9 | 1 |
| 701 | 281.4 | 0 | 146.8 | 52.2 | 1 |
| 701 | 288.4 | 0 | 147.7 | 54.5 | 1 |
| 701 | 297.8 | 0 | 142.4 | 58.7 | 1 |
| 701 | 311.0 | 0 | 135.7 | 63.2 | 1 |
| 701 | 320.2 | 0 | 129.9 | 61.7 | 1 |

For the latter two, multiple Tethyan ocean basins opened and closed between Gondwana and the China blocks. The China blocks only became part of Eurasia in the latest Jurassic or earliest Cretaceous [97,164], adding further paleogeographic complexity, as summarized below (Fig 3).

A continental realm dubbed 'Greater Adria' [50,165], roughly the size of Greenland, occupied much of the area that intervened Africa and Eurasia in the Mediterranean realm. Greater Adria broke off northern Gondwana (Africa), where it occupied the region between the east Tunisian and Levant margins during the Triassic-Jurassic opening of the Eastern Mediterranean ocean. It separated from Iberia and Eurasia by the opening of the Alpine Tethys ocean that was linked to the opening of the Atlantic Ocean [166,167], and it was bounded in the northeast by the Neotethys Ocean. The latter

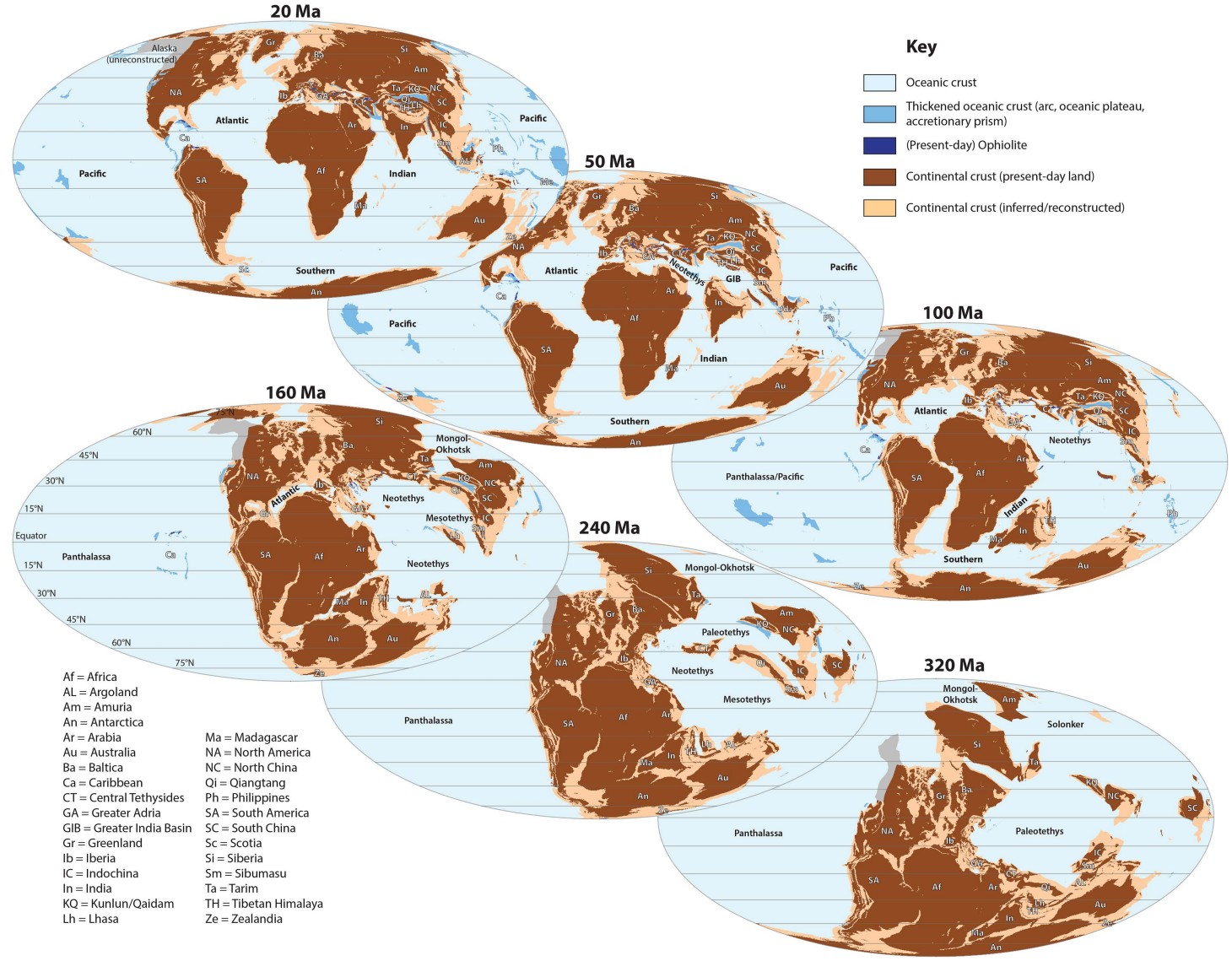

**Fig 3. Global paleogeography snapshots of the Utrecht Paleogeography Model that shows the distribution of continental and oceanic crust, placed in the paleomagnetic reference frame based on gAPWP25 [14] (Table 2).** The associated GPlates files are provided in the Supplementary information in S1 File (available in a DOI).

opened during Triassic to Jurassic time when ribbons of continental crust preserved in the Balkans and the Pontides and Lesser Caucasus broke off northern Gondwana. These were transferred to the Eurasian margin, closing the Paleotethys ocean in the north(east) and opening the Neotethys in their wake in the south(west) [50,168] (Fig 3). Such continental ribbons between a 'Paleotethys' and 'Neotethys' have been identified throughout the Tethyan realm and are referred to as 'Cimmerian continents' [169], but the ages of opening of Neotethys and Paleotethys vary between the segments identified above. Greater Adria was internally strongly extended and mostly submarine and was covered by limestones whose deformed remnants now make up the Apennines, southern Alps, Dinarides, Hellenides, and Anatolide-Tauride mountain belts [50,168,170–172] (Fig 3). The Alps, Carpathians, and eastern Balkans were derived from subducted Eurasian continental margin lithosphere [67,173,174].

To the east, the Iranian segment consists of a 'Cimmerian' microcontinental ribbon whose remains occupy much of Iran and north Afghanistan (Fig 3). However, Paleotethys closure and Neotethys opening on either side of this continent pre-dated the Cimmerian history of the Mediterranean region: the Iranian Cimmerian bock broke off the Arabian margin in the late Permian and collided with Eurasia in the Late Triassic [169,175–177]. In the Jurassic to Early Cretaceous, the Iranian Cimmerian block was broken in fragments by the opening of back-arc basins that subsequently closed in Late Cretaceous to Eocene time [87,178]. In the Iranian segment, the Neotethys was a few thousand kilometers wide and subducted from the Jurassic until the Oligocene onset of Arabia-Eurasia collision [88,179].

The Tibetan and Himalayan segment of the Neotethys has seen a series of microcontinents rifting off Gondwana and colliding with the North and South China Blocks (Fig 3). First, a continental ribbon rifted off Gondwana in the Late Carboniferous to Early Permian. This consisted of the Qiangtang terrane of Tibet (or terranes – some interpret multiple continental blocks that collided sometime in the late Paleozoic or early Mesozoic [180], which continued to the east (and at present, southeast) as the Sibumasu and west Sumatra terranes. The Indochina block, that presently occupies much of Thailand, Laos, Cambodia, and Vietnam, in turn broke off in this process from Sibumasu to open a narrow oceanic basin in its wake [181]. These blocks collided with South China and the Kunlun arc of northern Tibet, which was part of the North China Block, in the Late Triassic [98,181–183]. This process closed the Paleotethys Ocean to the north and opened the 'Mesotethys' Ocean to the South. Subsequently, the Lhasa terrane, which likely started rifting from the Greater Indian and west Australian margin of Gondwana in the Late Carboniferous [184], drifted northwards between Late Triassic and Early Cretaceous time [185]. This closed the Mesotethys and opened the Neotethys Ocean. Late Carboniferous-Early Permian and Late Triassic rifting also affected the western Australian margin [186] and separated continental fragments that finally broke off in the latest Jurassic to form 'Argoland' [187]. This microcontinental archipelago, together with the intra-oceanic Woyla arc, collided in Cretaceous to Eocene time with Sibumasu and West Sumatra [74,188]. During this process, in early Cretaceous time, rifting started within the Greater Indian margin of north Gondwana, reflected by a series of Lower Cretaceous rift-related volcanics found in the northern, 'Tibetan' Himalaya [189]. Paleomagnetic data and tectonic reconstructions show that the Tibetan Himalaya became separated from Greater India in the Cretaceous, and drifted northwards to close the Neotethys ocean and opening a 'Greater India Basin' in its wake [91] (Fig 3). This interpretation remains debated [190], but because our reconstruction everywhere systematically follows paleomagnetic evidence, the calculator does so for the Tibetan Himalayan terrane too. In the Early Cretaceous, also India broke off Gondwana and started its northward journey, leaving microcontinents in its wake (e.g., west of Australia [163,191] and the Seychelles [192]) due to ridge jumps in the Indian Ocean. The Greater India basin closed in the Eocene and Oligocene, after the Neotethys closed and Tibetan Himalaya collided with southern Tibet around 60 Ma [193] and until the arrival of the Indian continental margin in the latest Oligocene to middle Miocene [194].

The China Blocks to the north of the Tethyan oceans, however, were not part of Eurasia until the latest Jurassic or earliest Cretaceous, when the Mongol-Okhotsk Ocean closed [164]. This ocean opened in Permian time as a back-arc basin behind a subduction zone that consumed ocean floor of the western Panthalassa Ocean, and that broke a continental ribbon known as 'Amuria' from Siberia [164,195]. The Mongol-Okhotsk Ocean started closing again in the Late Triassic, when the North China Block collided with Amuria in the south forming the Solonker Suture. The North China Block had broken off eastern Gondwana in Devonian time [196], and gradually moved north until it collided with Amuria in the Late Triassic. Around that same time, the South China Block, that also broke off Gondwana in the Devonian [197], collided with North China [198]. After the Late Triassic, the China Blocks together with the Tibetan terranes became a single continent that moved north towards Siberia, becoming part of Eurasia following Mongol-Okhotsk closure (Fig 3).

## 6. New online interface and functionality

The Paleolatitude 3.0 online tool is available on www.paleolatitude.org and provides two options to compute paleolatitudes. On the home screen, any location may be chosen on the map by a mouse click, and a graph will appear showing

the paleolatitudinal evolution of that location since 320 Ma (or shorter, if the selected location is part of an oceanic plate or polygon that formed after 320 Ma). A maximum of ten curves may be computed at a time (Fig 4). The graph can be downloaded as various Fig formats, including as a vector image and the underlying calculated paleolatitudes and their uncertainties can be downloaded as an Excel file.

The home screen contains a button that opens a page with Advanced options. At the top, a list of previously mouse click-selected locations is provided, to which the user may manually add locations with a specified latitude and longitude, and if desired, a specific age or age range (Fig 4). In addition, the user may choose the preferred paleomagnetic reference frame. The default is the frame is Vaes et al. [14], with indication of the version of the underlying dataset. At the time of writing, this is version gAPWP25. Future updates will be indicated on the Paleolatitude.org website, and will be made available on the accompanying site www.apwp-online.org [103]. Alternatively, the user may select the reference frame based on older APWPs [21–23]. There is also an option for the user to modify the graph axes on the home screen (Fig 4).

Finally, the Advanced Options page offers a 'batch option', where the user may upload a data file for bulk paleolatitude computation. There is no maximum number of data, but very large data files (with 10.000s of entries) may take a few hours to compute. The bulk option requires an Excel or CSV file that provides input information on the location, name, and age of the samples, and the desired reference frame (Fig 4).

## 7. Comparison with other models

We illustrate the use of the new batch option in Paleolatitude.org 3.0 through a comparison with a recently published dataset of tetrapod dinosaurs and their paleogeographic distribution from the Permian to the Cretaceous [199]. This study compiled the data from the global paleobiology database [200]. That database provides a paleolatitude for each of its entries based on reference frames of Scotese and coworkers for older entries [201,202] or of Wright et al. [203] using a spline-fitted paleomagnetic reference frame of Torsvik and van der Voo [204] for younger entries. However, Heath et al. [199], preferring a more recent plate model and paleomagnetic reference frame, recalculated the paleolatitudes using the GPlates reconstruction of Merdith et al. [205] placed in the paleomagnetic reference frame of Tetley [206].

The relative plate motions between the major plates bounded by modern oceans vary little between these different reconstructions, except the China Blocks before the Cretaceous, which vary strongly between models (compare Fig 5e with the rest of the curves). Most differences in predicted paleolatitude arise from the different reference frames used (Fig 5). The Scotese reconstructions use a hybrid of a hotpot reference frame and paleomagnetic reference frame. The reference frame used in Wright et al. [203] corresponds to the 60–550 Ma APWP for Gondwana of Torsvik and van der Voo [204]. This APWP was computed from paleopoles of Gondwanan continents only, using a spherical spline approach in which the paleopoles are weighted based on a set of seven criteria for pole quality (Q-factor [207]). This approach provides a highly smoothed reference frame that does not come with a quantified uncertainty. Because this APWP is not defined for the past 60 Ma, any paleolatitude computed using this APWP is based on an interpolation for almost the entire Cenozoic. This APWP was therefore not intended to serve as a global paleomagnetic reference frame. The same team published updated Gondwanan APWPs, as well as global APWP that also included data from, e.g., North America and Eurasia, twice afterward, superseding the original Torsvik and van der Voo work [21,208].

The (unpublished) paleomagnetic reference frame of Tetley [206] uses the paleopole database of Torsvik et al. [21], but 'optimizes' the APWP by an algorithm that aims to minimize both absolute plate velocities as well as their gradients. This algorithm uses an iterative process in which the position and age of each paleopole is allowed to freely change within their errors bounds, until the APWP converges to a path that jointly minimizes the mean APW velocity and velocity gradient. Because this approach does not strictly follow the data, and because the underlying data are averages of poles that in turn are arbitrary collections of data [30], this paleomagnetic reference frame does not come with a geologically meaningful error bar. The version of the paleomagnetic reference frame of Tetley [206] used in the Merdith et al. [205] plate model was constructed using a running mean approach with a 50 Myr time window, leading to enhanced smoothing compared to

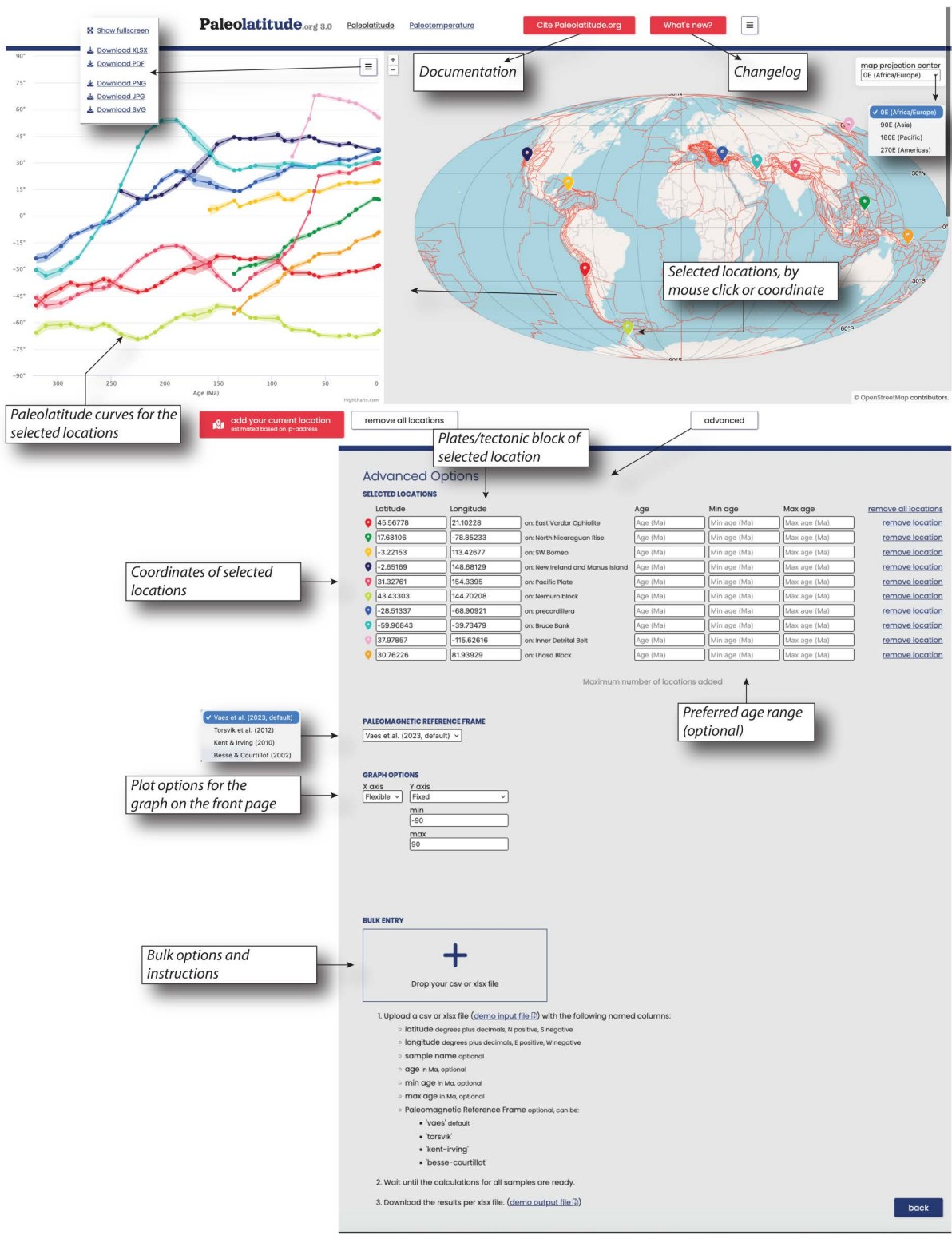

**Fig 4. Outline of the Paleolatitude.org 3.0 web interface, and advanced options, including a bulk data option with instructions.**

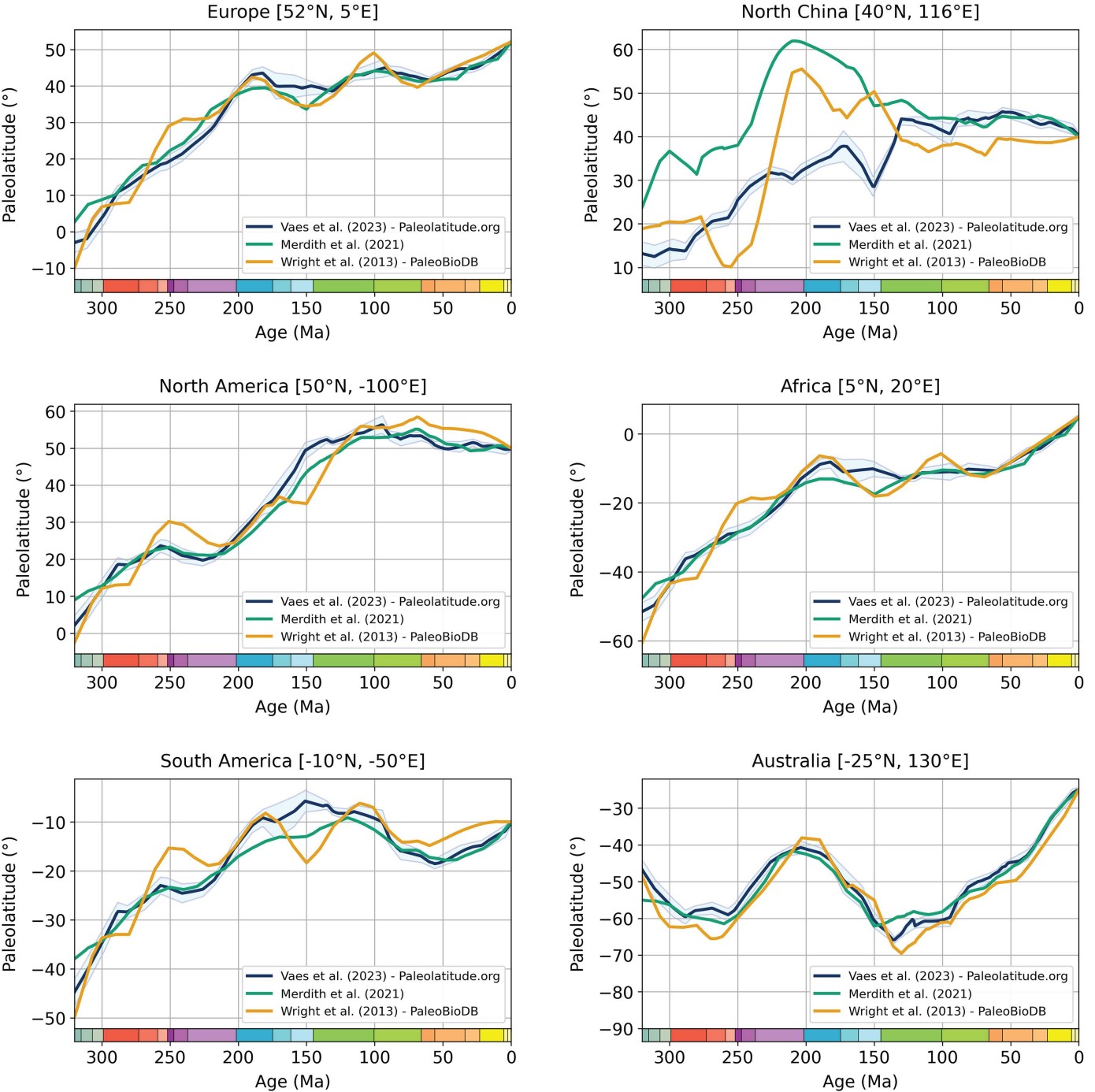

**Fig 5. Paleolatitude curves for coordinates in selected major continents illustrating the differences between widely used paleogeographic models.**

the Torsvik et al. [21] global APWP for 320−0 Ma. As a result of this approach, polar wander rates were reduced by ~56% compared to pole-averaged running mean paths. However, there is no rationale why polar wander rates must be minimal, and the approach thus smears and averages peaks that may well be signals of paleogeographic change.

We compare the paleolatitudes computed by Heath et al. [199] based on the Merdith et al. reconstruction with those from Paleolatitude 3.0 (Fig 6a). First, it is clear that the first-order distribution of tetrapod dinosaurs across latitudes that underpinned the interpretations of Heath et al. [199], is robust. However, we may use this dataset to illustrate the differences of the Paleolatitude.org 3.0 model with the other two widely used models. The Merdith et al. and Tetley models [205,206] give paleolatitudes that are systematically more northerly by up to ~10° between ~270 and 210 Ma, and more southerly by up to ~5° after this time. This difference illustrates the effects of the 50 Myr sliding window and the smoothing optimization approach used by Tetley [206]. In addition, the China blocks in the reconstruction of Merdith et al. [205] are 20–25° farther north than in the Utrecht Paleogeography Model. Differences with the paleolatitudes given in the global paleobiology database are on the same order of magnitude, although these predict latitudes that are systematically more northerly (Fig 6c), by up to 10°.

The differences between these different reconstructions do not change first-order distribution estimates (as illustrated with the Heath et al. [199] dataset, Fig 6), although they may be meaningful for critical intervals such as the paleo-polar circle or paleo-tropics. More importantly, the Paleolatitude.org calculator provides uncertainties that are a function of the error in the paleomagnetic reference frame and the age range assigned to the sample. This opens the opportunity to propagate these uncertainties into quantitative estimates of distributions, for instance in biodiversity gradients, as illustrated below.

## 8. Application: Propagating uncertainty in biodiversity gradients

The Latitudinal Diversity Gradient (LDG) is a macroecological pattern of higher taxonomic richness at lower than at at higher latitudes – and more so in marine organisms than in terrestrial ones – that is thought to result from higher and less variable solar irradiance at lower latitudes [209,210]. The LDG has been sensitive to climatic processes, such as the steepness of the latitudinal temperature gradient or hyperthermal events leading to low-latitude diversity crises [211,212]. The LDG is computed from fossil occurrence data placed in temporal and paleogeographic context. With the Paleolatitude.org tool, it is now for the first time possible to not only determine for each fossil its paleolatitude at its median age, as has so far been the common approach, but also to include the effects of age and paleolatitudinal uncertainty.

Here we illustrate the use of Paleolatitude.org 3.0 with an example of a collection of ~34,000 Upper Jurassic marine fossils. From these, we calculated the LDG using the Paleobiology Database accessed using the paleobioDB package for R Software [213]. Occurrences for marine fauna identified at least at the genus level were downloaded and uncertain genus identifications were culled. All occurrences whose stratigraphic range overlapped with Late Jurassic and recorded at any spatial resolution were included in our collection. Based on each occurrence's current geographical location and age range, its paleogeographic position was determined with the Paleolatitude.org tool and the paleolatitudinal uncertainty was determined from the combination of the uncertainty in the paleomagnetic reference frame and the age uncertainty of each fossil, as illustrated in Fig 1d. Note that of our dataset, approximately 1000 fossils came from Alaska and the Canadian Cordillera (Fig 7a) that have not been included in our reconstruction yet. We discarded these data, which would have occupied low- to mid- northern hemisphere paleolatitudes [23,156]. We stress that the low to mid-latitude LDG in our example is thus likely somewhat underestimated.

The average uncertainty, defined as the difference between the higher and lower bound, for the Upper Jurassic marine genera is 6.8° (Fig 7b), mostly as a function of age uncertainty. We applied no cutoffs or data curation (which may be advisable when carrying out an in-depth LDG study) but used all data for further analysis.

We performed an analysis of Sampled In Bin (SIB) richness gradient and accounted for the uncertainty in paleolatitude using a bootstrap approach. To this end, paleolatitudes were divided into 5° bins. When an occurrence's paleogeographic uncertainty range spanned multiple bins, we calculated the proportion of its latitudinal range falling within each overlapping bin.

In each bootstrap iteration (n = 1000), every occurrence was assigned to one of its overlapping bins, with frequency based on the calculated proportions. For each iteration, we counted the number of unique genera per bin (SIB). The final

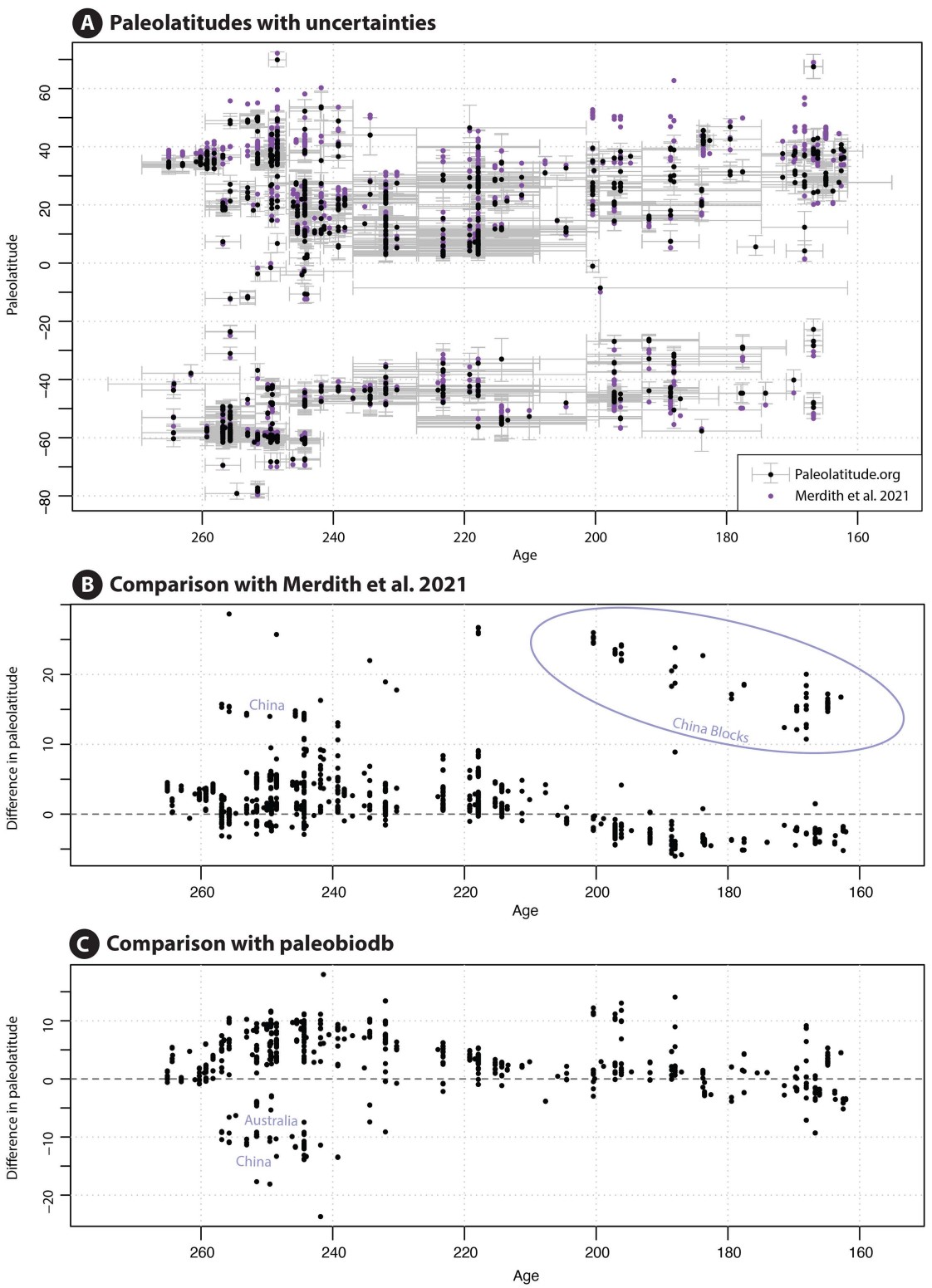

**Fig 6. Differences in paleolatitude estimates for an example dataset of tetrapod dinosaurs found in Upper Permian to Middle Jurassic strata** [199]. **A)** Distribution of data according to the global paleobiology database [200] that uses reference frames of Scotese and coworkers [201,202] or of Wright et al. [203] using a spline-fitted paleomagnetic reference frame of Torsvik and van der Voo [204]; **B)** Distribution of data using the reconstruction

of Merdith et al. [205] in the unpublished optimized paleomagnetic reference frame of Tetley [206]; **C)** Data distribution using our new paleogeographic reconstruction in the upgraded gAPWP25 [14]; **D)** Difference between A and C; **E)** Difference between B and **C.**

richness estimate for each bin represents the mean SIB across all iterations, with 95% confidence intervals calculated from the bootstrap distribution (Fig 8). For comparison, we also calculated SIB using point-estimate paleolatitudes only, assigning each occurrence to a single bin based on its estimated paleolatitude without accounting for positional uncertainty (Fig 8). In most cases, the uncertainty-corrected richness estimates overlap with point-estimates, reflecting the high precision of reconstructed paleolatitudes. Mean SIB richness with uncertainty accounted for is slightly higher than point estimate across the −45° to 45° interval, with the largest difference in the 35°-40° bin, where richness peaks. This is mostly the result of taking age uncertainty into account, which spreads fossils over a wider range of bins than just the bin of their median age. Summarizing, our example based on a large dataset of tens of thousands of samples quantitatively corroborates the robustness of LDG calculations in which uncertainties were previously not considered. As illustrated in Fig 8, uncertainties for smaller datasets may be considerably higher and may affect previous conclusions based on the semi-quantitative, error bar-less paleolatitude calculations that have so far been the standard.

## 9. Conclusions

In this paper, we provide an upgrade of the Paleolatitude.org webtool to version 3.0. This tool provides estimates of paleolatitude through time for any location on Earth and computes a paleolatitudinal uncertainty that is a function of the underlying paleogeographic reconstruction and the age uncertainty of a sample. The new features include the following:

1) We provide the first global model, back to ~320 Ma, that restores the paleogeographic units that are now thrusted over each other in orogenic (mountain) belts and provide the underlying GPlates reconstruction files. In addition, we provide a brief synopsis of global paleogeography since the Carboniferous, particularly including the formation and demise by collision of microcontinents that existed in the Tethyan, and to a lesser extent, the Panthalassa/Paleo-Pacific Oceans.

2) We place this reconstruction into a recent, more precise paleomagnetic reference frame that is based on site-level paleomagnetic data. In this paper, we provide the first update of its underlying database, increasing the database by ~10% and further decreasing uncertainty.

3) We introduce a new online interface with an easy-to-use tool with a batch option that allows computing paleolatitudinal data for essentially unlimited datasets.

4) Finally, we illustrate differences with previous reconstructions and explain these differences. We show an application by calculating a paleolatitudinal biodiversity gradient for the late Jurassic in which we use a bootstrap approach to propagate paleolatitude and age uncertainty into the result.

## Supporting information

**S1 Data.** Supplementary Files: A set of supplementary files is available at DOI: https://doi.org/10.6084/m9.figshare.31021144, containing the following elements: S1 File. GPlates files (www.gplates.org [15]) of the Utrecht Paleogeography Model presented in the paleogeographic maps of Figs 1 and 3, and the rigid polygon version that is used as basis for the Paleolatitude.org tool. These consist of a rotation file, and a series of shape files (in gpml format) that underpin the paleogeographic model, a gpml file of the rigid polygons that are used to rotate coordinates in the Paleolatitude.org tool, as well as a project file (gproj) of the entire paleogeographic reconstruction. S2 File. Details of gAPWP25. In addition to a Readme.txt file with general descriptions and the gAPWP25.rot file with the updated paleomagnetic

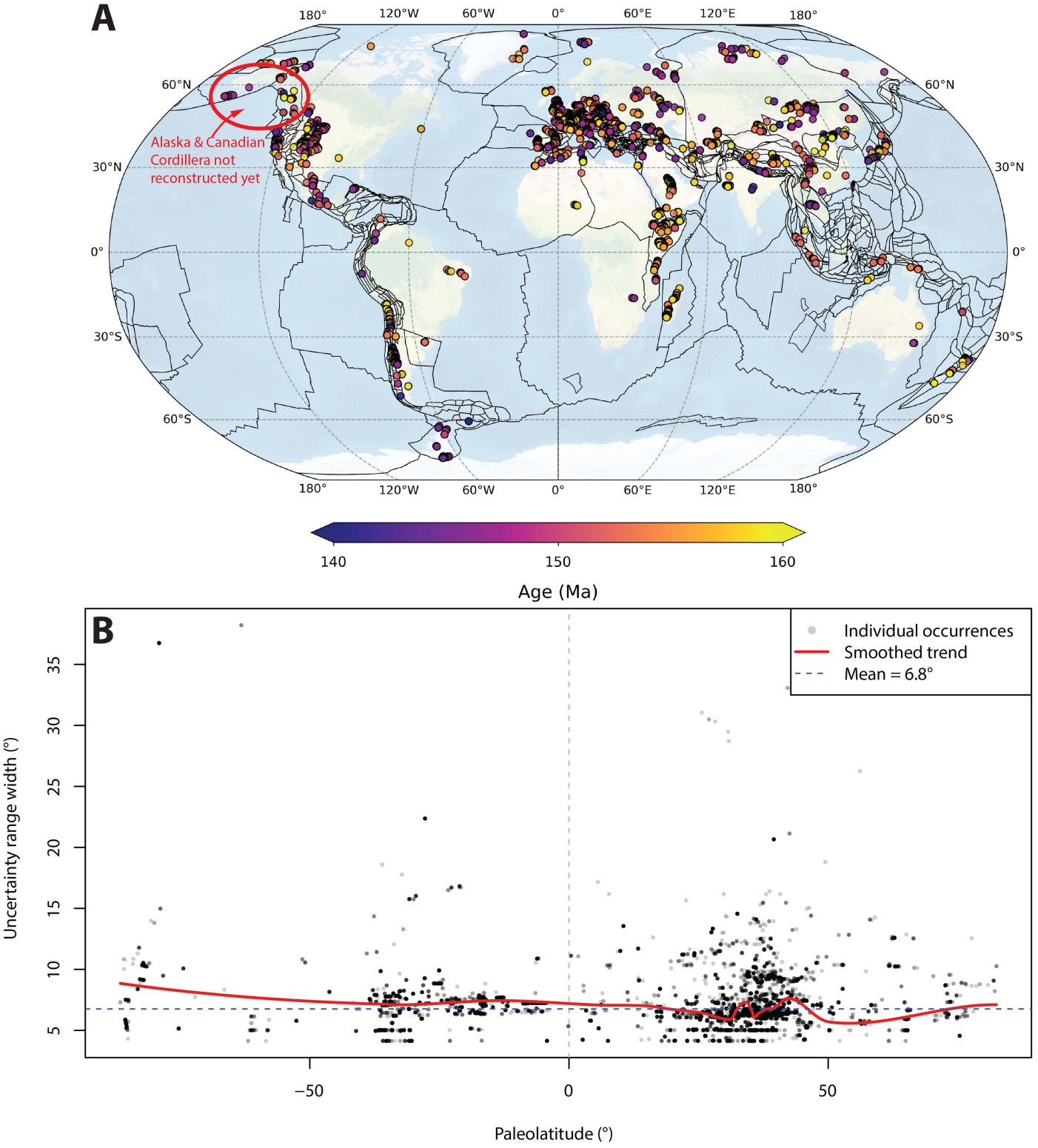

**Fig 7. A) Geographic distribution of the marine fossil dataset from the Upper Jurassic used for to compute a Latitudinal Diversity Gradient. B)** Paleolatitude precision as a function of paleolatitude. Points indicate individual occurrences, with darker gray indicating more overlying observations. The red line is Loess fit through the data. Based on 33803 observations.

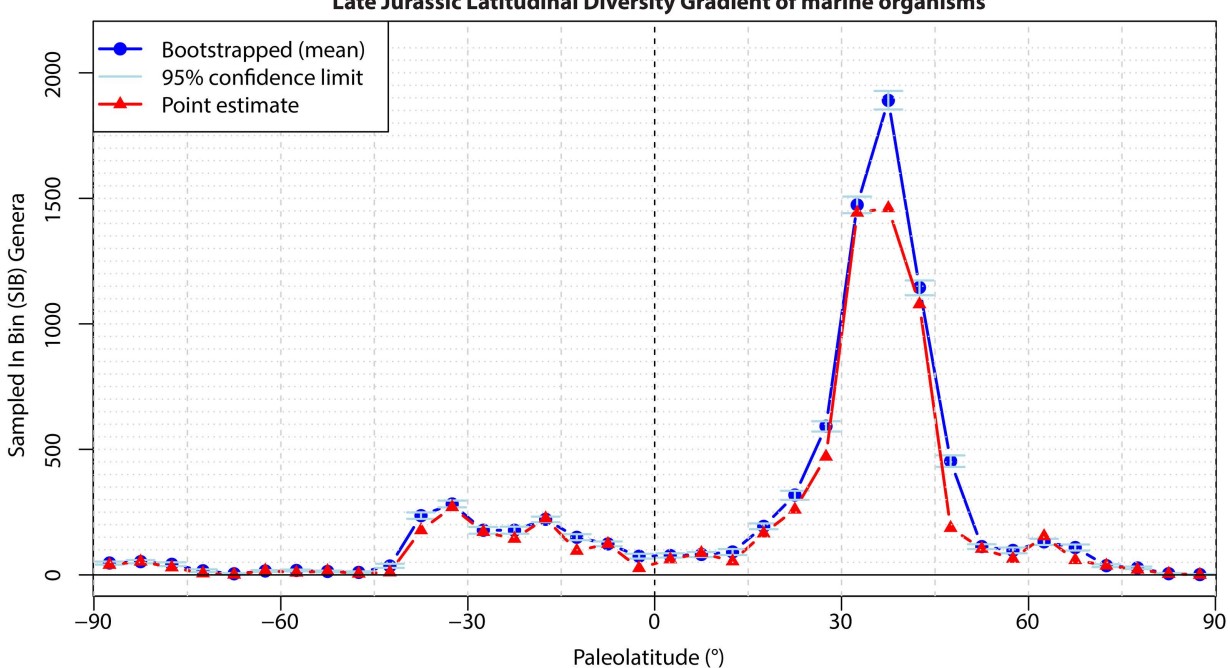

**Fig 8. Genus-level Latitudinal Diversity Gradient of marine organisms in the Late Jurassic, without curation, and taking uncertainty in age and the paleomagnetic reference frame into account when computing paleolatitude, reflected in 95% bootstrap confidence intervals.** Overlain is a Sampled In Bin point estimate of the LDG in which paleolatitude and age uncertainty is not considered. Based on 33802 occurrences resampled 1000 times into 5° paleolatitude bins (see Supplementary Data (available in a DOI)).

reference frame in GPlates rotation format, the files contain: S1 Table. Changelog of the update of gAPWP23 to gAPWP25. S2 Table. Paleomagnetic database used to compute the global apparent polar wander path for the last 320 Ma. We have listed age constraints, statistical parameters, Euler rotation parameters and other metadata per paleo-magnetic pole used in the parametric re-sampling scheme. For more details, see main text. The grey-colored entries are excluded from the computation of the APWP. See columns 'age constraints', 'comments' and 'reliability' for specific details for a given dataset. Abbreviations: min_age and max_age = lower and upper boundaries of age uncertainty range; slat/slon = latitude and longitude of (mean) sampling location; N = number of paleomagnetic sites used to compute the paleopole; mDec/mInc = mean declination of inclination; α95/A95 = radius of the 95% confidence circle about the mean of the distribution of directions/VGPs; k/K = Fisher [109] precision parameter of the distribution of directions/VGPs; plat/plon = paleopole latitude and longitude (south pole); K_est/A95_est = values estimated using formula of Cox [214] (eq. 24); plateID = plate identification number; Rlat/Rlon = paleopole latitude and longitude in coordinate frame of South Africa; EP_lat/EP_lon/EP_ang = total reconstruction pole parameters for rotating the paleopole to South Africa coordinates; f = flattening factor (only for sedimentary data), p_std = standard deviation of the assumed normal distributed co-latitudes, obtained from E/I correction (only for sedimentary data); Deenen = indicates whether the N-dependent reliability envelope of Deenen et al. [215] is satisfied (TRUE or FALSE) or, in case of sediment-derived datasets, the quality grade (A, B or C) following the evaluation scheme of Vaes et al. [106]; excl = reason for exclusion (R = rejected because entry is a dupli-cate, N < 5, age range > 20 Ma, remagnetized or otherwise considered unreliable, see comments/reliability column); refno = reference number in global paleomagnetic database [216, 217]; DB = database in which entry is listed (T12 [21], PSV10 [218], gAPWP23 [14], gAPWP25 = added in this study). S3 Table. Global plate circuit used to transfer paleomagnetic data to a single reference plate, from Vaes et al. [14] with minor modification in the rotation parameters for India. See Vaes

et al. [14] for references and details. S4 Table. gAPWP25 rotated in the coordinates of the major continents. S3 File. S5 Table. Euler rotations of every polygon relative to South Africa (701) at times corresponding to the ages of the reference poles of gAPWP25, using the rotation file of the Utrecht Paleogeography Model in Supplementary Files 1. Paleolatitudes provided by Paleolatitude.org 3.0 are interpolated from these rotations.
(ZIP)

## Acknowledgments

M. Carrano, M. Clapham, T. Danelian, H. Eichiner, F. Fürsich, M. Gahr, D. Hempfling, A. Hendy, S. Hicks, W. Kiessling, A. Kocsis, K. Layou, E. Link, J. Martinelli, U. Merkel, S. Nürnberg, W. Puijk, P. Schossleitner, L. Villier contributed the largest part of fossil occurrences to the database used in this study. No permits were required for the described study, which complied with all relevant regulations. We thank Nicolas Flament and two anonymous reviewers for their comments.

## Author contributions

**Conceptualization:** Douwe J. J. van Hinsbergen.

**Formal analysis:** Douwe J. J. van Hinsbergen, Bram Vaes.

**Funding acquisition:** Douwe J. J. van Hinsbergen.

**Investigation:** Douwe J.J. van Hinsbergen, Bram Vaes, Lydian M. Boschman, Nalan Lom, Suzanna H. A. van de Lagemaat, Eldert L. Advokaat, Sanne de Baar, Emilia B. Jarochowska.

**Methodology:** Douwe J. J. van Hinsbergen, Bram Vaes, Emilia B. Jarochowska.

**Project administration:** Douwe J. J. van Hinsbergen.

**Resources:** Douwe J. J. van Hinsbergen.

**Software:** Menno R. T. Fraters, Joren Paridaens.

**Supervision:** Douwe J. J. van Hinsbergen.

**Validation:** Douwe J. J. van Hinsbergen.

**Visualization:** Douwe J. J. van Hinsbergen, Lydian M. Boschman, Emilia B. Jarochowska.

**Writing – original draft:** Douwe J. J. van Hinsbergen, Bram Vaes, Emilia B. Jarochowska.

**Writing – review & editing:** Douwe J. J. van Hinsbergen, Bram Vaes, Lydian M. Boschman, Nalan Lom, Suzanna H. A. van de Lagemaat, Eldert L. Advokaat, Sanne de Baar, Emilia B. Jarochowska.

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
