## [Decision Letter · Decision Letter 0]

24 Feb 2026

Dear Dr. van Hinsbergen,

Thank you for submitting your manuscript to PLOS ONE. After careful consideration, we feel that it has merit but does not fully meet PLOS ONE’s publication criteria as it currently stands. Therefore, we invite you to submit a revised version of the manuscript that addresses the points raised during the review process.

We look forward to receiving your revised manuscript.

Kind regards,

Carlo Meloro

Academic Editor

PLOS One

Journal Requirements:

Additional Editor Comments (if provided):

This is a good paper quite rich in text. I recommend streamlining the longest sections maybe using sub-headings. I found particularly hard to follow the:

Section 3. plate and orogen reconstruction approach;

Section 5. A brief synopsis of global paleogeography since the Carboniferous

Please reduce text.

As highlighted also by one reviewer the doi figshare file provided in line 743 does not work. Please fix it.

Reviewers' comments:

Reviewer's Responses to Questions

**Comments to the Author**

1. Is the manuscript technically sound, and do the data support the conclusions?

Reviewer #1: Yes

Reviewer #2: Yes

Reviewer #3: Yes

2. Has the statistical analysis been performed appropriately and rigorously?

Reviewer #1: Yes

Reviewer #2: Yes

Reviewer #3: Yes

3. Have the authors made all data underlying the findings in their manuscript fully available?

Reviewer #1: Yes

Reviewer #2: No

Reviewer #3: Yes

4. Is the manuscript presented in an intelligible fashion and written in standard English?

Reviewer #1: Yes

Reviewer #2: Yes

Reviewer #3: Yes

Reviewer #1: Dear authors, it is very interesting work, however there are lots of language and editorial problems throughout the manuscript. 1. Therefore, would you please edit from the abstract to the references?

2. use the researchers in place of 'we'

3.

Reviewer #2: This manuscript will be well cited because it accompanies the release of an update to https://paleolatitude.org/. My comments about the format (missing tables and glitches with supplementary files) should be straightforward to address. I made some suggestions for the authors to consider.

L. 46: 'wholesale rotation of the solid Earth (crust and mantle)' – with respect to what?

L. 63: version 2.0 is mentioned further on; consider mentioning it in the introduction to set the scene.

L. 157 and elsewhere: ref [13] is published but the citation is to a preprint

L. 509: 'This interpretation remains debated [190]'. Many details of section '5. A brief synopsis of global paleogeography since the Carboniferous' (L. 370–529 and Figures 1 and 3) could be subject to debate. One block that caught my eye is the South China sea, which I would place further to the west. This point is not relevant in the context of latitude, but it illustrates that I paid attention to this block because I have worked on it (Young et al., 2019; Marks et al., 2025). I anticipate that other readers may be looking carefully at regions that they are interested in. To alleviate this problem, consider opening the section by mentioning that choices/interpretations have to be made in a global model and that the model is openly available for anyone to modify. In other words, the model (as any model) is likely to be incorrect in some aspects, but it remains useful.

L. 587: 'The (unpublished) paleomagnetic reference frame of Tetley [205]': the reference frame by Tetley et al. (2019) is not a paleomagnetic reference frame, and it is published (https://agupubs.onlinelibrary.wiley.com/doi/full/10.1029/2019JB017442;
https://zenodo.org/records/2638121). The starting point of that reference frame was 'the paleopole database of Torsvik et al. [27]' (L. 587–588). The motivation was to produce a 'geodynamically reasonable' reference frame. Müller et al. (2022) built on the work of Tetley et al. (2019) and called their optimised reference frame a 'mantle reference frame'. Consider making the point somewhere that different reference frames are suitable for different applications: paleomagnetic reference frames for surface observations; mantle reference frames (no-net-rotation, optimised) for mantle processes; and TPW-corrected reference frames for processes related to the evolution of Earth's core.

Fig. 2a: it would be helfpul to explain the relationship between global apparent polar wander path and true polar wander somewhere in the manuscript. The two are presented on the same level in Figure 2a without explanation. True polar wander is briefly explained in the text, but the concept of global apparent polar wander path and its relevance to reference frames is not introduced.

I could find Tables 1, 2 or 3 anywhere in the manuscript.

The provided link to the data did not work: https://doi:10.6084/m9.figshare.31021144. I managed to find this link: https://figshare.com/articles/online_resource/Supplementary_Information_to_van_Hinsbergen_et_al_PLoS_One_submitted_2026_/31021144?file=60861682.

The GPlates project points to a cpt file that is not included (see screenshot).

There is a problem with the packaging of the files in the GPlates project provided in supplement: the rotations would not apply to the data in the project (see screenshots at 160 Ma and 320 Ma with projections similar to that used in Figures 1 and 3).

Nicolas Flament, Wollongong, 19 February 2026

Reviewer #3: This research paper entitled “Paleolatitude.org 3.0: a calculator for paleoclimate and paleobiology studies based on a new global paleogeography model,” presents updates to the Paleolatitude.org calculator. These updates include a global paleogeographic model with GPlates reconstruction files extending back to 320 Ma, restoration of paleogeographic units in orogenic belts, and a more precise paleomagnetic reference frame with updated statistical procedures. A new, user-friendly interface supports batch processing and data export. Additionally, to illustrate differences with previous reconstructions, the paper demonstrates the calculation of a Late Jurassic paleolatitudinal biodiversity gradient using a bootstrap approach to propagate uncertainties in paleolatitude and age. These improvements enhance the tool's accuracy and usability, and I recommend publication with minor revisions.

My main comments are as follows:

1. Line 6: The position of the corresponding author should be marked with an asterisk (*).

2. Line 19: The corresponding author's email should be written according to the journal's requirements: * d.j.j.vanhinsbergen@uu.nl.

3. Lines 48–49: For -oceanography and -biology, it is recommended to use the full words.

4. Line 50: Change [2-5] to [2–5]. It is recommended to replace the hyphen (-) in the references in the text with the en dash (–) as required by the journal format.

5. Lines 50–53: I recommended to rewrite this sentence.

6. Lines 97, 362, 387: Change “Figure 1:”, “Figure 2:”, “Figure 3.”, and “Figure 4.” to “Fig 1.”, “Fig 2.”, “Fig 3.”, and “Fig 4.”.

7. Line 323: Change “Table 1:” to “Table 1.”.

8. Lines 181–182: “Tibetan Plateau” is recommended to be written as “Tibetan Plateau (Qinghai-Xizang Plateau)”, and “Tien Shan” is recommended to be written as “Tian Shan”.

9. In the text, figure titles should be abbreviated: for example, "Figure 3" should be "Fig 3"; "Fig. 8" should be "Fig 8"; "Figures 1 and 3" should be "Figs 1 and 3". The journal PLOS One uses "Fig X." for figure titles, and "Figure" in the main text should be abbreviated to "Fig".

10. The supplementary tables S1–S5 appear not to be cited in the main text.

11. Line 607: Change “Figure 6a” to “Fig 6A”; Line 618: Change “Figure 6c” to “Fig 6C”; Line 659: Change “Figure 7a” to “Fig 7A”; Line 664: Change “Figure 7b” to “Fig 7B”; Line 658: Change “Figure 1d” to “Fig 1D”. The Figs 1A, 1B and 1C; Fig 6B are not referenced.

12. I recommended to standardize the spelling of certain words throughout the text, for example, "paleo-" versus "palaeo-".

13. I recommended to standardize the labeling letters (e.g., a, b, c) under all images in the article, using either all uppercase or all lowercase consistently to avoid mixing cases.

14. Please check the formatting of the main text to ensure it complies with the journal's requirements.

15. Please check the citation format of all references; for those with DOI, include the DOI link: references with DOI should include "https://doi.. ..." at the beginning.

.

Reviewer #1: No

Reviewer #2: **Yes:**Nicolas FlamentNicolas FlamentNicolas FlamentNicolas Flament

Reviewer #3: No

---

## [Author Response · Author response to Decision Letter 1]

2 Mar 2026

Dear editor, please find the explanation of our edits in the rebuttal letter below, in blue.

Additional Editor Comments (if provided):

This is a good paper quite rich in text. I recommend streamlining the longest sections maybe using sub-headings. I found particularly hard to follow the:

Section 3. plate and orogen reconstruction approach;

Section 5. A brief synopsis of global paleogeography since the Carboniferous

We added subheadings.

Please reduce text.

As highlighted also by one reviewer the doi figshare file provided in line 743 does not work. Please fix it.

Apologies. The doi has been corrected.

Reviewers' comments:

Reviewer's Responses to Questions

Comments to the Author

3. Have the authors made all data underlying the findings in their manuscript fully available?

Reviewer #1: Yes

Reviewer #2: No

Reviewer #3: Yes

The doi of the Figshare link has been corrected.

5. Review Comments to the Author

Reviewer #1: Dear authors, it is very interesting work, however there are lots of language and editorial problems throughout the manuscript. 1. Therefore, would you please edit from the abstract to the references?

2. use the researchers in place of 'we'

We have re-read the manuscript and corrected the errors where we found them (but they were rare).

It is quite clear that 'we' refers to the authors, and is less wordy. Hence, we prefer to keep the style as we had it.

Reviewer #2: This manuscript will be well cited because it accompanies the release of an update to https://paleolatitude.org/. My comments about the format (missing tables and glitches with supplementary files) should be straightforward to address. I made some suggestions for the authors to consider.

L. 46: 'wholesale rotation of the solid Earth (crust and mantle)' – with respect to what?

Thanks for spotting. We added 'relative to the spin axis'.

L. 63: version 2.0 is mentioned further on; consider mentioning it in the introduction to set the scene.

Done. We added to the introduction ' timescales). ...the online paleolatitude calculator of Paleolatitude.org was developed about a decade ago [6], was shortly after updated to version 2.0 that included the Paleozoic [7], and has since become...etc.

L. 157 and elsewhere: ref [13] is published but the citation is to a preprint

Thanks for spotting. We corrected the doi.

L. 509: 'This interpretation remains debated [190]'. Many details of section '5. A brief synopsis of global paleogeography since the Carboniferous' (L. 370–529 and Figures 1 and 3) could be subject to debate. One block that caught my eye is the South China sea, which I would place further to the west. This point is not relevant in the context of latitude, but it illustrates that I paid attention to this block because I have worked on it (Young et al., 2019; Marks et al., 2025). I anticipate that other readers may be looking carefully at regions that they are interested in. To alleviate this problem, consider opening the section by mentioning that choices/interpretations have to be made in a global model and that the model is openly available for anyone to modify. In other words, the model (as any model) is likely to be incorrect in some aspects, but it remains useful.

We added a general disclaimer to the opening of section 5, as suggested:

"As with any global paleogeographic model, it is inevitable that our version of it made choices that are under debate and scrutiny. Below, we describe the general characteristics of the model, and the sources that were used for the compilation. All GPlates reconstruction files are available in the supplementary information, openly available for anyone to modify."

South China Sea: the model shows the distribution of continental and oceanic crust (as noted in the caption of Figure 3). The oceanic crust of the South China Sea is precisely where our model indicates it, it doesn't depend on any model choices on our side. The extensional basin, and the sea itself, continues farther west of course.

L. 587: 'The (unpublished) paleomagnetic reference frame of Tetley [205]': the reference frame by Tetley et al. (2019) is not a paleomagnetic reference frame, and it is published (https://agupubs.onlinelibrary.wiley.com/doi/full/10.1029/2019JB017442;
https://zenodo.org/records/2638121). The starting point of that reference frame was 'the paleopole database of Torsvik et al. [27]' (L. 587–588). The motivation was to produce a 'geodynamically reasonable' reference frame. Müller et al. (2022) built on the work of Tetley et al. (2019) and called their optimised reference frame a 'mantle reference frame'. Consider making the point somewhere that different reference frames are suitable for different applications: paleomagnetic reference frames for surface observations; mantle reference frames (no-net-rotation, optimised) for mantle processes; and TPW-corrected reference frames for processes related to the evolution of Earth's core.

Actually, that is not true. The Merdith et al reconstruction used by Heath did not use the published mantle reference frame of Tetley (and shouldn't indeed, for that frame cannot see true polar wander) but used the unpublished paleomagnetic frame that Tetley produced in his PhD thesis.

Fig. 2a: it would be helfpul to explain the relationship between global apparent polar wander path and true polar wander somewhere in the manuscript. The two are presented on the same level in Figure 2a without explanation. True polar wander is briefly explained in the text, but the concept of global apparent polar wander path and its relevance to reference frames is not introduced.

We added as opening to section 2 (Methods...):

Paleogeographic models are developed from relative global plate tectonic reconstructions (e.g., ref [16]), placed in a frame that positions the plates, oceans, and continents relative to a chosen, 'fixed' reference. There are two types of such reference frames: the ones where plates are positioned relative to the mantle (mantle reference frames [17-20]), or relative to the Earth's magnetic field that on geological times aligns with the spin axis (paleomagnetic reference frames [14, 21-23]. Mantle reference frames only 'see' motion of plates relative to the mantle, but not the common rotation of mantle and plates together relative to the spin axis (i.e., true polar wander). However, true polar wander may significantly change the distribution of global geography relative to the equator and poles, and is must therefore be included in paleogeographic reconstructions. Therefore, it is of importance that paleo-climate, -environment, or -biology is studied in paleogeographic context placed in the paleomagnetic reference frame, for only that frame provides quantitative information of the paleolatitude that determines the angle of solar insolation [6].

I could find Tables 1, 2 or 3 anywhere in the manuscript.

We double checked the resubmission to make sure that the tables are properly included.

The provided link to the data did not work: https://doi:10.6084/m9.figshare.31021144. I managed to find this link: https://figshare.com/articles/online_resource/Supplementary_Information_to_van_Hinsbergen_et_al_PLoS_One_submitted_2026_/31021144?file=60861682.

We corrected the link

The GPlates project points to a cpt file that is not included (see screenshot).

There is a problem with the packaging of the files in the GPlates project provided in supplement: the rotations would not apply to the data in the project (see screenshots at 160 Ma and 320 Ma with projections similar to that used in Figures 1 and 3).

Thanks for spotting! We corrected this and made sure the supplementary files & the figshare files were updated.

Nicolas Flament, Wollongong, 19 February 2026

Reviewer #3: This research paper entitled “Paleolatitude.org 3.0: a calculator for paleoclimate and paleobiology studies based on a new global paleogeography model,” presents updates to the Paleolatitude.org calculator. These updates include a global paleogeographic model with GPlates reconstruction files extending back to 320 Ma, restoration of paleogeographic units in orogenic belts, and a more precise paleomagnetic reference frame with updated statistical procedures. A new, user-friendly interface supports batch processing and data export. Additionally, to illustrate differences with previous reconstructions, the paper demonstrates the calculation of a Late Jurassic paleolatitudinal biodiversity gradient using a bootstrap approach to propagate uncertainties in paleolatitude and age. These improvements enhance the tool's accuracy and usability, and I recommend publication with minor revisions.

My main comments are as follows:

1. Line 6: The position of the corresponding author should be marked with an asterisk (*).

Added.

2. Line 19: The corresponding author's email should be written according to the journal's requirements: * d.j.j.vanhinsbergen@uu.nl.

Done

3. Lines 48–49: For -oceanography and -biology, it is recommended to use the full words.

Corrected throughout the text

4. Line 50: Change [2-5] to [2–5]. It is recommended to replace the hyphen (-) in the references in the text with the en dash (–) as required by the journal format.

Corrected throughout the text

5. Lines 50–53: I recommended to rewrite this sentence.

We do not see what is unclear about the sentence.

6. Lines 97, 362, 387: Change “Figure 1:”, “Figure 2:”, “Figure 3.”, and “Figure 4.” to “Fig 1.”, “Fig 2.”, “Fig 3.”, and “Fig 4.”.

Corrected throughout the text

7. Line 323: Change “Table 1:” to “Table 1.”.

Corrected throughout the text.

8. Lines 181–182: “Tibetan Plateau” is recommended to be written as “Tibetan Plateau (Qinghai-Xizang Plateau)”, and “Tien Shan” is recommended to be written as “Tian Shan”.

We believe Tibetan Plateau is clear enough. We replaced Tien Shan with Tian Shan.

9. In the text, figure titles should be abbreviated: for example, "Figure 3" should be "Fig 3"; "Fig. 8" should be "Fig 8"; "Figures 1 and 3" should be "Figs 1 and 3". The journal PLOS One uses "Fig X." for figure titles, and "Figure" in the main text should be abbreviated to "Fig".

Corrected.

10. The supplementary tables S1–S5 appear not to be cited in the main text.

Correct.

11. Line 607: Change “Figure 6a” to “Fig 6A”; Line 618: Change “Figure 6c” to “Fig 6C”; Line 659: Change “Figure 7a” to “Fig 7A”; Line 664: Change “Figure 7b” to “Fig 7B”; Line 658: Change “Figure 1d” to “Fig 1D”. The Figs 1A, 1B and 1C; Fig 6B are not referenced.

Corrections made. Not every subfigure is cited in the text, but they are relevant where they were referred to where referenced to the general figure.

12. I recommended to standardize the spelling of certain words throughout the text, for example, "paleo-" versus "palaeo-".

Palaeo was only used in the reference list, where used by the original authors.

13. I recommended to standardize the labeling letters (e.g., a, b, c) under all images in the article, using either all uppercase or all lowercase consistently to avoid mixing cases.

They were. A, B, C is used for subfigures, a, b, c for references to citated literature.

14. Please check the formatting of the main text to ensure it complies with the journal's requirements.

Done.

15. Please check the citation format of all references; for those with DOI, include the DOI link: references with DOI should include "https://doi.. ..." at the beginning.

We followed the formats provided by the original sources.

---

## [Decision Letter · Decision Letter 1]

24 Mar 2026

Paleolatitude.org 3.0: a calculator for paleoclimate and paleobiology studies based on a new global paleogeography model

PONE-D-26-01713R1

Dear Dr. van Hinsbergen,

We’re pleased to inform you that your manuscript has been judged scientifically suitable for publication and will be formally accepted for publication once it meets all outstanding technical requirements.

Kind regards,

Carlo Meloro

Academic Editor

PLOS One

Additional Editor Comments (optional):

All good, please ensure to incorporate one of reviewers minor point and update the right figshare link.

Reviewers' comments:

Reviewer's Responses to Questions

**Comments to the Author**

Reviewer #2: (No Response)

Reviewer #3: All comments have been addressed

2. Is the manuscript technically sound, and do the data support the conclusions?

Reviewer #2: Yes

Reviewer #3: Yes

3. Has the statistical analysis been performed appropriately and rigorously?

Reviewer #2: I Don't Know

Reviewer #3: Yes

4. Have the authors made all data underlying the findings in their manuscript fully available?

Reviewer #2: Yes

Reviewer #3: Yes

5. Is the manuscript presented in an intelligible fashion and written in standard English?

Reviewer #2: Yes

Reviewer #3: Yes

Reviewer #2: All but one of my comments have been addressed.

It would be helfpul to explain the relationship between global apparent polar wander path and true polar wander somewhere in the manuscript (perhaps the first time the concept of global apparent polar wander path is introduced?).

Some text has been added to provide some background about reference frames and about global tectonic reconstructions. I think these are good additions.

The tables are now there, the links have been fixed and the GPlates rotation file now works.

I had assumed the reference was to Tetley et al. (2019), but I can now see that it was to Mike Tetley's PhD thesis.

I had meant to comment on the (longitudinal) location of the South China block during early Permian times, but had forgotten to mention the period of interest. This comment is not important in the context of paleolatitudes.

Nicolas Flament, Wollongong, 4 March 2026

Reviewer #3: The authors made detailed revisions based on the reviewers' suggestions. I suggest publishing this manuscript.

.

Reviewer #2: **Yes:**Nicolas FlamentNicolas FlamentNicolas FlamentNicolas Flament

Reviewer #3: No

---

## [Editor Report · Acceptance letter]

PONE-D-26-01713R1

PLOS One

Dear Dr. van Hinsbergen,

I'm pleased to inform you that your manuscript has been deemed suitable for publication in PLOS One. Congratulations! Your manuscript is now being handed over to our production team.

Kind regards,

on behalf of

Dr. Carlo Meloro

Academic Editor

PLOS One